# AutoFHE: Automated Adaption of CNNs for Efficient Evaluation over FHE

## Abstract

Secure inference of deep convolutional neural networks (CNNs) was recently demonstrated under the fully homomorphic encryption (FHE) scheme, specifically the Full Residue Number system variant of Cheon-Kim-Kim-Song (RNS-CKKS). The state-of-the-art solution uses a high-order composite polynomial to approximate non-arithmetic ReLUs and refreshes zero-level ciphertext through bootstrapping. However, this solution suffers from prohibitively high latency, both due to the number of levels consumed by the polynomials (47%) and the inference time consumed by bootstrapping operations (70%). Furthermore, it requires a hand-crafted architecture for homomorphically evaluating CNNs by placing a bootstrapping operation after every Conv-BN layer. To accelerate CNNs on FHE and automatically design a homomorphic evaluation architecture, we propose *AutoFHE: Automated adaption of CNNs for evaluation over FHE*. AutoFHE exploits the varying sensitivity of approximate activations across different layers in a network and jointly evolves polynomial activations (EvoReLUs) and searches for placement of bootstrapping operations for evaluation under RNS-CKKS. The salient features of AutoFHE include: i) a multi-objective coevolutionary (MOCoEv) search algorithm to maximize validation accuracy and minimize the number of bootstrapping operations, ii) a gradient-free search algorithm, R-CCDE, to optimize EvoReLU coefficients, and iii) polynomial-aware training (PAT) to fine-tune polynomial-only CNNs for a few epochs to adapt trainable weights to EvoReLUs. We demonstrate the efficacy of AutoFHE through the evaluation of ResNets on encrypted CIFAR-10 and CIFAR-100 under RNS-CKKS. Experimental results on CIFAR-10 indicate that in comparison to the state-of-the-art solution, AutoFHE can reduce inference time (50 images on 50 threads) by up to 3,297 seconds (43%) while preserving the accuracy (92.68%). AutoFHE also improves the accuracy of ResNet-32 on CIFAR-10 by 0.48% while accelerating inference by 382 seconds (7%).

## 1 Introduction

Fully homomorphic encryption (FHE) is a promising solution for secure inference of neural networks (Gilad-Bachrach et al., 2016; Brutzkus et al., 2019; Lou & Jiang, 2021; Lee et al., 2022b;a). However, Homomorphically evaluating CNNs on encrypted data is challenging in two respects: 1) the design of homomorphic evaluation architecture of deep CNNs with arbitrary depth and 2) non-arithmetic operations like ReLU. Recently, FHE-MP-CNN (Lee et al., 2022a) successfully implemented a homomorphic evaluation architecture of ResNets by using bootstrapping (Cheon et al., 2018a; Bossuat et al., 2021) to refresh zero-level ciphertext under the full residue number system (RNS) variant of Cheon-Kim-Kim-Song (RNS-CKKS) scheme (Cheon et al., 2017; 2018b). However, since FHE supports only homomorphic

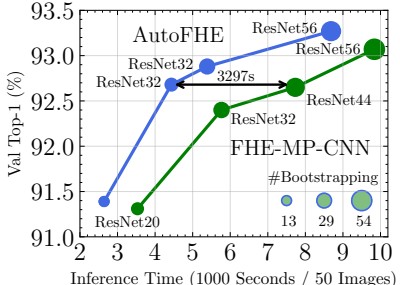

Figure 1: Pareto fronts of AutoFHE *versus* FHE-MP-CNN on encrypted CIFAR-10 under the RNS-CKKS FHE scheme.

multiplication and addition, non-arithmetic operations are approximated by polynomials (Gilad-Bachrach et al., 2016; Chou et al., 2018; Brutzkus et al., 2019; Lee et al., 2021a;c; 2022a). For example, FHE-MP-CNN adopts a high-precision Minimax composite polynomial (Lee et al., 2021a;c)

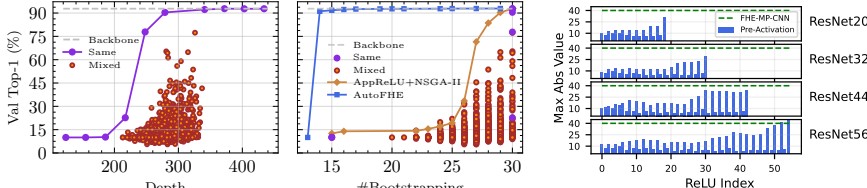

Figure 2: **Motivating AutoFHE.** Left: depth consumption of AppReLUs based on ResNet-20 backbone on CIFAR-10. The purple line is when the same precision AppReLU is used in all layers, while the red circles show 5000 randomly-sampled combinations of mixed-precision layerwise AppReLUs. Middle: the number of bootstrapping operations where we show trade-offs of the same AppReLU and mixed AppReLUs as in the left panel. We also show a multi-objective search result using mixed-precision layerwise AppReLUs and the Pareto front of the proposed AutoFHE. Right: distributions of pre-activations (the maximum absolute values) of ResNets on CIFAR-10 where the green line corresponds to $B$, the scale value of AppReLU in FHE-MP-CNN.

with degree $\{15, 15, 27\}$ to approximate ReLUs (AppReLU). A more comprehensive discussion of related work is in Appendix B.

FHE-MP-CNN, the state-of-the-art approach, is limited by three main design choices. *First*, high-precision approximations like AppReLU only consider function-level approximation and neglect the potential for end-to-end optimization of the entire network response. As such, the same high-precision AppReLU is used to replace all the network's ReLU layers, which necessitates the evaluation of very deep circuits. *Secondly*, due to the high number of levels required for each AppReLU, ciphertexts encrypted with leveled HE schemes like CKKS quickly exhaust their levels. Therefore, a bootstrapping operation is necessary for each AppReLU to refresh the level of zero-level ciphertexts. While these design choices are collectively very effective at maintaining the performance of the plaintext networks under FHE, they require many multiplicative levels and, consequently, numerous bootstrapping operations. *Thirdly*, due to the constraints imposed by the cryptographic scheme (RNS-CKKS in this case), inference of networks in FHE requires the codesign of AppReLU and the homomorphic evaluation architecture. These include the careful design of AppReLU (number of composite polynomials and their degrees), cryptographic parameters, placement of bootstrapping operations, and choice of network architectures to evaluate.

We illustrate the limitations of FHE-MP-CNN's design choices through a case study (Figure 2) of ResNet-20 on CIFAR-10. We consider two plausible solutions to trade-off accuracy and computational burden of FHE-MP-CNN. (i) **Same Precision AppReLU:** We replace all ReLU layers with AppReLU of a given precision. We can trade-off (purple line in the left panel) accuracy and depth consumption using AppReLU with different precision. However, as the middle panel shows, these solutions (purple dots) do not necessarily translate to a trade-off between accuracy and the number of bootstrapping operations due to many wasted levels. All the trade-off solutions collapse to either 15 or 30 bootstrapping operations. (ii) **Mixed-Precision AppReLU:** Each ReLU layer in the network can be replaced by AppReLU of *any* precision. We randomly sample 5,000 combinations of mixed-precision layerwise AppReLUs and show (red dots) their depth consumption and the number of bootstrapping operations in the left and middle panels, respectively. Observe that layerwise mixed-precision AppReLU leads to a better trade-off between accuracy and the number of bootstrapping operations. However, FHE-MP-CNN neglects the layerwise sensitivity (range) of ReLU pre-activations (the right panel shows the distribution of the layerwise maximum absolute value of pre-activation) and uses AppReLU which is optimized for a ReLU with a large pre-activation range. Therefore, the Pareto front of mixed-precision layerwise AppReLU optimized by a multi-objective search algorithm NSGA-II Deb et al. (2002) is still inferior to AutoFHE, our proposed solution, by a significant margin. In summary, while both the solutions we considered were able to reduce the number of bootstrapping operations, unlike AutoFHE, it also lead to severe loss in performance.

In this paper, we relax the design choices of FHE-MP-CNN and accelerate the inference of CNNs over homomorphically encrypted data while maximizing performance. The main premise behind our approach is to *directly optimize the end-to-end function represented by the network instead of optimizing the function represented by the activation function*. This idea allows us to exploit the varying sensitivity of activation function approximation across different layers in a network. Therefore, theoretically, evolving layerwise polynomial approximations of ReLUs (EvoReLU) should reduce the total multiplicative depth required by the resulting polynomial-only networks, and thus the number of time-consuming bootstrapping operations and the inference time on encrypted data. To

this end, we propose AutoFHE, a search-driven approach to jointly optimize layerwise polynomial approximations of ReLU and the placement of bootstrapping operations. Specifically, we propose a multi-objective co-evolutionary (MOCoEv) algorithm that seeks to *maximize* accuracy while simultaneously *minimizing* the number of bootstrapping operations. AutoFHE jointly searches for the parameters of the approximate activation functions at all layers, i.e., *degrees* and *coefficients* and the optimal placement of the bootstrapping operations in the network. Our contributions are three-fold:

1. AutoFHE automatically searches for EvoReLUs and bootstrapping operations. It provides a diverse set of Pareto-effective solutions that span the trade-off between accuracy and inference time under RNS-CKKS.

2. From an algorithmic perspective,

    (a) We propose a simple yet effective multi-objective co-evolutionary (MOCoEv) algorithm to effectively explore and optimize over the large search space ($10^{79} \sim 10^{230}$) and optimize high-dimensional vectors ($114 \sim 330$) corresponding to our formulation.

    (b) We design a gradient-free algorithm, regularized co-operative co-evolutionary differentiable evolution (R-CCDE), to *optimize* the coefficients of high-degree composite polynomials.

    (c) We introduce polynomial-aware training (PAT) to finetune EvoReLU DNNs for *a few* epochs.

3. Experimental results (Figure 1) on encrypted CIFAR-10 and CIFAR-100 under RNS-CKKS show that, compared to FHE-MP-CNN, the state-of-the-art approach, AutoFHE can effectively trade-off accuracy and inference time and result in a better Pareto front. On CIFAR-10, AutoFHE reduces inference time (50 images on 50 threads) by up to 3,297 seconds (43%) while preserving the accuracy (92.68%). Specifically, AutoFHE reduces inference time of ResNet-20, ResNet-32 (21 bootstrapping operations) and ResNet-56 by 25%, 23% and 12%, respectively, while improving accuracy up to 0.28%. AutoFHE also improves accuracy of ResNet-32 (29 bootstrapping operations) on CIFAR-10 by 0.48% while accelerating inference by 382 seconds (7%). On CIFAR-100, AutoFHE saves inference time by 972 seconds (17%) while preserving accuracy.

## 2 PRELIMINARIES

**RNS-CKKS:** The full residue number system (RNS) variant of Cheon-Kim-Kim-Song (RNS-CKKS) (Cheon et al., 2017; 2018b) is a leveled homomorphic encryption (HE) scheme for approximate arithmetic. Under RNS-CKKS, a ciphertext $\boldsymbol{c} \in \mathcal{R}_{Q_\ell}^2$ satisfies the decryption circuit $[\langle \boldsymbol{c}, sk \rangle]_{Q_\ell} = m + e$, where $\langle \cdot, \cdot \rangle$ is the dot product and $[\cdot]_Q$ is the modular reduction function. $\mathcal{R}_{Q_\ell} = \mathbb{Z}_{Q_\ell}[X]/(X^N + 1)$ is the residue cyclotomic polynomial ring. The modulus is $Q_\ell = \prod_{i=0}^{\ell} q_\ell$, where $0 \le \ell \le L$. $\ell$ is a non-negative integer referred to as *level*, and it denotes the capacity of homomorphic multiplications. $sk$ is the secret key with Hamming weight $h$. $m$ is the original plaintext message, and $e$ is a small error that provides security. A ciphertext has $N/2$ slots to accommodate $N/2$ complex or real numbers. RNS-CKKS supports homomorphic addition and multiplication:

$$\text{Homomorphic Addition: } \text{Decrypt}(\boldsymbol{c} \oplus \boldsymbol{c}') = \text{Decrypt}(\boldsymbol{c}) + \text{Decrypt}(\boldsymbol{c}') \approx m + m'$$
$$\text{Homomorphic Multiplication: } \text{Decrypt}(\boldsymbol{c} \otimes \boldsymbol{c}') = \text{Decrypt}(\boldsymbol{c}) \times \text{Decrypt}(\boldsymbol{c}') \approx m \times m' \quad (1)$$

**Bootstrapping:** Leveled HE only allows a finite number of homomorphic multiplications, with each multiplication consuming one level due to rescaling. Once a ciphertext's level reaches zero, a bootstrapping operation is required to refresh it to a higher level and allow more multiplications. The number of levels needed to evaluate a circuit is known as its *depth*. RNS-CKKS with bootstrapping (Cheon et al., 2018a) is an FHE scheme that can evaluate circuits of arbitrary depth. It enables us to homomorphically evaluate deep CNNs on encrypted data. Conceptually, bootstrapping homomorphically evaluates the decryption circuit and raises the modulus from $Q_0$ to $Q_L$ by using the isomorphism $\mathcal{R}_{q_0} \cong \mathcal{R}_{q_0} \times \mathcal{R}_{q_1} \times \cdots \times \mathcal{R}_{q_L}$ (Bossuat et al., 2021). Practically, bootstrapping (Cheon et al., 2018a) homomorphically evaluates modular reduction $[\cdot]_Q$ by first approximating it by a scaled sine function, which is further approximated through polynomials (Cheon et al., 2018a; Lee et al., 2020). Bootstrapping Bossuat et al. (2021) has four stages, including ModRaise, CoeffToSlot, EvalMod, and SlotToCoeff. These operations involve a lot of homomorphic multiplications and rotations, both of which are costly operations, especially the latter. The refreshed ciphertext has level $\ell = L - K$, where $K$ levels are consumed by bootstrapping (Bossuat et al., 2021) for polynomial approximation of modular reduction.

**FHE-MP-CNN (Lee et al., 2022a)** is the state-of-the-art framework for homomorphically evaluating deep CNNs on encrypted data under RNS-CKKS with high accuracy. Its salient features include 1) *Compact Packing:* All channels of a tensor are packed into a single ciphertext. Multiplexed parallel (MP) convolution was proposed to process the ciphertext efficiently. 2) *Homomorphic Evaluation Architecture:* Bootstrapping operations are placed after every Conv-BN, except for the first one, to refresh zero-level ciphertexts. This hand-crafted homomorphic evaluation architecture for ResNets is determined by the choice of cryptographic parameters, the level consumption of operations, and ResNet's architecture. 3) *AppReLU:* It replaces all ReLUs with the same high-order Minimax composite polynomial Lee et al. (2021a;c) of degrees $\{15, 15, 27\}$. By noting that $\text{ReLU}(x) = x \cdot (0.5 + 0.5 \cdot \text{sgn}(x))$, where $\text{sgn}(x)$ is the sign function, the approximated ReLU (AppReLU) is modeled as $\text{AppReLU}(x) = x \cdot (0.5 + 0.5 \cdot p_\alpha(x)), x \in [-1, 1]$. $p_\alpha(x)$ is the composite Minimax polynomial. The precision $\alpha$ is defined as $|p_\alpha(x) - \text{sgn}(\text{x})| \leq 2^{-\alpha}$. AppReLU is expanded to arbitrary domains $x \in [-B, B]$ of pre-activations in CNNs by scaling it as $B \cdot \text{AppReLU}(x/B)$. However, this reduces approximation precision to $B \cdot 2^{-\alpha}$. To estimate the maximum dynamic range $B$ (40 for CIFAR-10 and 65 for CIFAR-100) of ReLUs, FHE-MP-CNN evaluates the pre-trained network on the training dataset. FHE-MP-CNN uses the same dynamic range $B$ for all polynomials and neglects the uneven distribution of pre-activations as shown in Figure 2. Explicitly accounting for this uneven distribution allows us to use smaller $B'$ and $\alpha'$ but with the same precision, i.e., $B' \cdot 2^{-\alpha'} = B \cdot 2^{-\alpha}$, for $B' < B$ and $\alpha' < \alpha$. 4) *Cryptographic Parameters:* FHE-MP-CNN sets $N = 2^{16}$, $L = 30$ and Hamming weight $h = 192$. Please refer to Lee et al. (2022a) for the detailed implementation of FHE-MP-CNN and other parameters. These parameters provide 128-bits of security Cheon et al. (2019). 5) *Depth Consumption:* To reduce level consumption, FHE-MP-CNN integrates scaling parameter $B$ into Conv-BN. The multiplicative depth consumption of Bootstrapping (i.e., $K$), AppReLU, Conv, DownSampling, AvgPool, FC and BN layers are 14, 14, 2, 1, 1, 1, 0, respectively. Statistically, when using FHE-MP-CNN to homomorphically evaluate ResNet-18/32/44/56 on CIFAR-10 or CIFAR-100, AppReLUs consume $\sim 47\%$ of total levels and bootstrapping operations consume $\sim 70\%$ of inference time.

## 3 AutoFHE: Joint EvoReLU and Bootstrapping Search

To minimize the latency of secure inference dominated by bootstrapping operations induced by high-degree polynomials and automatically design suitable homomorphic evaluation architecture, we propose AutoFHE. It is designed to search for layerwise polynomial approximation of ReLU jointly with the placement of bootstrapping. Furthermore, we directly optimize the end-to-end objective to facilitate finding the optimal combination of layerwise polynomials.

### 3.1 EvoReLU

**EvoReLU** is defined as $y = \text{EvoReLU}(x) = x \cdot \left(0.5 + p^d(x)\right), x \in [-1, 1], y \in [0, 1]$. The composite polynomial $p^d(x) = (p_K^{d_K} \circ \cdots \circ p_k^{d_k} \circ \cdots \circ p_1^{d_1})(x), 1 \leq k \leq K$ approximates $0.5 \cdot \text{sgn}(x)$. The composite polynomial $p^d(x)$ has $K$ sub-polynomial functions and degree $d = \prod_{k=1}^{K}$. This structure for EvoReLU bears similarity to the Minimax composite polynomial in Lee et al. (2021c; 2022a). However, the objective for optimizing the coefficients is significantly different. We represent the composite polynomial $p^d(x)$ by its degree vector $\boldsymbol{d} = \{d_i\}_{i=1}^{d_K}$, and each sub-polynomial $p_k^{d_k}(x)$ as a linear combination of Chebyshev polynomials of degree $d_k$, i.e., $p_k^{d_k}(x) = \beta_k \sum_{i=1}^{d_k} \alpha_i \text{T}_i(x)$, where $\text{T}_i(x)$ are the Chebyshev bases of the first kind, $\alpha_i$ are the coefficients for linear combination and $\beta_k$ is a parameter to scale the output. The coefficients $\boldsymbol{\alpha}_k = \{\alpha_i\}_{i=1}^{d_k}$ control the polynomial's shape, while $\beta_k$ controls its amplitude. $\boldsymbol{\lambda} = (\boldsymbol{\alpha}_1, \beta_1, \cdots, \boldsymbol{\alpha}_k, \beta_k, \cdots, \boldsymbol{\alpha}_K, \beta_K)$ are the learnable parameters of EvoReLU with the degree $\boldsymbol{d}$.

**Homomorphic Evaluation Architecture:** The ResNet architecture comprises two types of connections, a chain, and a residual connection (see Fig. 3). To extend the domain of EvoReLU from $[-1, 1]$ to $[-B, B]$ but avoid extra depth consumption for scaling, we scale the plaintext weight and bias of BatchNorm by $1/B$ in advance for chain connections. But for residual connections, we cannot integrate the scale $1/B$ into BatchNorm's weight and bias. In this case, we scale the ciphertext output of the residual connection by $1/B$ at the expense of one level. Finally, we integrate $B$ into coefficients of $p_K^{d_K}(x)$ to re-scale the output of EvoReLU by $B$. Given the pre-activation $x \in [-B, B]$, the scaled

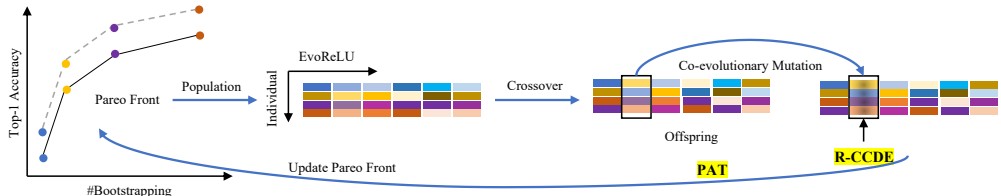

Figure 3: Homomorphic evaluation architectures of the chain and residual connections. Upper: standard ResNet Conv-BN-ReLU triplet He et al. (2016). Middle: FHE-MP-CNN. Bottom: AutoFHE, where dashed rectangles are placement of bootstrapping layers to search.

Figure 4: Overview of Multi-objective co-evolutionary (MOCoEv) search algorithm.

EvoReLU with the degree $d$ is parameterized by $\boldsymbol{\lambda}$:

$$y = \text{EvoReLU}(x, \boldsymbol{\lambda}; \boldsymbol{d}) = x \cdot (0.5 + p^d(x)), \text{ where } x \in [-B, B], y \in [0, B] \qquad (2)$$

where we estimate $B$ values for layerwise EvoReLUs on the training dataset. From Figure 3, FHE-MP-CNN places bootstrapping after every Conv-BN, while AutoFHE will search for placement of bootstrapping operations by adapting to different depth consumption of layerwise EvoReLUs.

**The Depth Consumption** of EvoReLU is $1 + \sum_{k=1}^{K} \lceil \log_2(d_k + 1) \rceil$ when using the Baby-Step Giant-Step (BSGS) algorithm Lee et al. (2020); Bossuat et al. (2021) to evaluate $p^d(x)$.

## 3.2 MOCoEv: Multi-Objective Co-Evolutionary Search

**Search Objectives:** Given a neural network function $f$ with $L$ ReLUs and the pre-trained weights $\boldsymbol{\omega}_0$, our goal is to maximize the accuracy of the network while minimizing its inference latency on encrypted data. A possible solution to achieve this goal is to maximize validation accuracy while minimizing the total multiplicative depth of the network with EvoReLUs. However, this solution does not practically accelerate inference since bootstrapping contributes most to latency, and this solution may not necessarily lead to fewer bootstrapping operations. Therefore, we optimize the parameters of all the EvoReLUs to maximize accuracy and directly seek to minimize the number of bootstrapping layers through a multi-objective optimization problem:

$$\min_{\boldsymbol{D}} \quad \{1 - \text{Acc}_{val}(f(\boldsymbol{\omega}^*); \boldsymbol{\Lambda}^*(\boldsymbol{D}), \boldsymbol{D}), \text{Boot}(\boldsymbol{D})\}$$
$$\text{s.t.} \quad \boldsymbol{\Lambda}^* = \arg\max_{\boldsymbol{\Lambda}} \{\text{Acc}_{val}(f(\boldsymbol{\omega_0}); \boldsymbol{\Lambda}(\boldsymbol{D}), \boldsymbol{D})\} \qquad (3)$$
$$\boldsymbol{\omega}^* = \arg\min_{\boldsymbol{\omega}} \mathcal{L}_{train}(f(\boldsymbol{\omega}); \boldsymbol{\Lambda}^*(\boldsymbol{D}), \boldsymbol{D})$$

where $\text{Acc}_{val}$ is the Top-1 accuracy on a validation dataset $val$, Boot is the number of bootstrapping operations, $\boldsymbol{D} = \{\boldsymbol{d}_1, \boldsymbol{d}_2, \cdots, \boldsymbol{d}_L\}$ is the degree vector of all EvoReLUs, the corresponding parameters are $\boldsymbol{\Lambda} = \{\boldsymbol{\lambda}_1, \boldsymbol{\lambda}_2, \cdots, \boldsymbol{\lambda}_L\}$, $f(\boldsymbol{\omega}_0)$ is the neural network with the pre-trained weights $\boldsymbol{\omega}_0$, $\mathcal{L}_{train}$ is the training loss. Given a degree vector $\boldsymbol{D}$, the number and placement of bootstrapping operations can be deterministically determined. Given $\boldsymbol{D}$, we can optimize $\boldsymbol{\Lambda}$ to maximize the validation accuracy. We further fine-tune the network $f(\cdot)$ to minimize the training loss $\mathcal{L}_{train}$. The objectives in equation 3 guide the search algorithm to, i) explore layerwise EvoReLU including its *degrees* and *coefficients*; 2) discover the placement of bootstrapping to work well with EvoReLU; 3) trade-off between validation accuracy and inference speed to return a diverse set of Pareto-effective solutions. In this paper, we *propose MOCoEv* to optimize the multi-objective $\min_{\boldsymbol{D}}\{1 - \text{Acc}_{val}(f(\boldsymbol{\omega}^*); \boldsymbol{\Lambda}^*(\boldsymbol{D}), \boldsymbol{D}), \text{Boot}(\boldsymbol{D})\}$. We *propose R-CCDE* and use an evolutionary criterion to maximize $\text{Acc}_{val}(f(\boldsymbol{\omega}_0); \boldsymbol{\Lambda}(\boldsymbol{D}), \boldsymbol{D})$. We *propose PAT* to fine-tune approximated networks with EvoReLUs to minimize $\mathcal{L}_{train}(f(\boldsymbol{\omega}); \boldsymbol{\Lambda}^*(\boldsymbol{D}), \boldsymbol{D})$.

**Search Space:** Our search space includes the number of sub-polynomials ($K$) in our composite polynomial, choice of degrees for each sub-polynomial ($d_k$) and the coefficients of the polynomials $\mathbf{\Lambda}$. Table 1a shows the options for each of these variables. Note that choice $d_k = 0$ corresponds to an identity placeholder, so theoretically, the composite polynomial may have fewer than $K$ sub-polynomials. Furthermore, when the degree of $(p_k^{d_k} \circ p_{k-1}^{d_{k-1}})(x)$ less than or equal to 31 (maximum degree of a polynomial supported on RNS-CKKS Lee et al. (2021a;c)), we merge the two sub-polynomials into a single sub-polynomial $p_k^{d_k}(p_{k-1}^{d_{k-1}})(x)$ with degree $d_k \cdot d_{k-1} \leq 31$ before computing its depth. This helps reduce the size of the search space and leads to smoother exploration. Tab. 1b lists the number of ReLUs of our backbone models and the corresponding dimension and size of search space for $\mathbf{D}$.

| Variable | Option |
|---|---|
| # polynomials ($K$) | 6 |
| poly degree ($d_k$) | $\{0, 1, 3, 5, 7\}$ |
| coefficients ($\mathbf{\Lambda}$) | $\mathbb{R}$ |

| Backbone | #ReLUs | Dimension of $\mathbf{D}$ | Search Space Size |
|---|---|---|---|
| ResNet-20 | 19 | 114 | $10^{79}$ |
| ResNet-32 | 31 | 186 | $10^{130}$ |
| ResNet-44 | 43 | 258 | $10^{180}$ |
| ResNet-56 | 55 | 330 | $10^{230}$ |

(a)             (b)

Table 1: (a) Search variables and options; (b) AutoFHE search space for ResNets.

**MOCoEv:** To overcome the challenge of multi-objective search over a high-dimensional $\mathbf{D}$ and explore the massive search space, we propose a multi-objective co-evolutionary (MOCoEv) search algorithm. Our approach is inspired by the *divide-and-conquer* strategy of cooperative co-evolution (CC) Yang et al. (2008); Mei et al. (2016); Ma et al. (2018). The key idea of MOCoEv is to decompose the *high*-dimensional multi-objective search problem to multiple *low*-dimensional sub-problems. MOCoEv includes i) *Decomposition:* given a Pareto-effective solution $\mathbf{D} = \{d_1, d_2, \cdots, d_L\}$, MOCoEv improves $\mathbf{D}$ by locally mutating $d_\ell, 1 \leq \ell \leq L$ so that $\mathbf{D}' = \{d_1, d_2, \cdots, d'_\ell, \cdots, d_L\}$ dominates $\mathbf{D} = \{d_1, d_2, \cdots, d_\ell, \cdots, d_L\}$ in terms of the validation accuracy and the number of bootstrapping; and ii) *Cooperative Evaluation:* we maintain the Pareto front as the context Mei et al. (2016) so we can evaluate the locally mutated solutions cooperatively with each other $d_j, j \neq \ell, 1 \leq j \leq L$. Figure 4 shows a step of one iteration of MOCoEv. During one iteration, we repeat the step $L$ times until we update all EvoReLUs. We design *crossover* and *co-evolutionary mutation* of MOCoEv to explore and exploit: (1) *Crossover:* given the current Pareto front, we select mating individuals to generate offspring and crossover offspring to exchange *genes* across EvoReLUs. For example, given two mating individuals $\mathbf{D}_1$ and $\mathbf{D}_2$, we crossover them to obtain $\mathbf{D}'_1 = \{b_\ell : b_\ell \in \mathbf{D}_1 \cup \mathbf{D}_2, 1 \leq \ell \leq L\}, \mathbf{D}'_2 = \{b_\ell : b_\ell \in (\mathbf{D}_1 \cup \mathbf{D}_2)/\mathbf{D}'_1, 1 \leq \ell \leq L\}$; (2) *Co-Evolutionary Mutation:* we mutate the $\ell$-th EvoReLU offspring, obtain a new Pareto front from mutated offspring and the current population, and finally update the current population. Then, we move onto the $(\ell + 1)$-th EvoReLU and repeat (1)(2) until $L$ EvoReLUs are updated. Therefore, we update the Pareto front $L$ times at the end of each iteration. We design three types of operators to mutate a composite polynomial function. i) randomly *replace* one polynomial sub-function with a new polynomial. ii) randomly *remove* a sub-function. iii) randomly *insert* a new polynomial. Please refer to Appendix C for background on evolutionary search algorithms. The implementation details of MOCoEv algorithm are in Appendix D.1.

### 3.3 REGULARIZED COOPERATIVE DIFFERENTIABLE CO-EVOLUTION

To solve $\mathbf{\Lambda}^* = \arg\max_{\mathbf{\Lambda}} \{\text{Acc}_{val}(f(\boldsymbol{\omega_0}); \mathbf{\Lambda}(\mathbf{D}), \mathbf{D})\}$ in equation 3 where $\mathbf{D} = \{d_\ell | 1 \leq \ell \leq L\}$, $\mathbf{\Lambda} = \{\boldsymbol{\lambda}_\ell | 1 \leq \ell \leq L\}$, we propose regularized cooperative co-evolutionary differentiable evolution (R-CCDE). Given degree $d_\ell$, it optimizes $\boldsymbol{\lambda}_\ell$ for function approximation level. However, the function approximation solution $\mathbf{\Lambda}$ maybe not the optimal solution for $\max_{\mathbf{\Lambda}} \{\text{Acc}_{val}(f(\boldsymbol{\omega}); \mathbf{\Lambda}(\mathbf{D}), \mathbf{D})\}$. So, we use MOCoEv to update the Pareto front in terms of the validation accuracy and the number of bootstrapping. R-CCDE decomposes $\boldsymbol{\lambda}_\ell$ into $\{\boldsymbol{\alpha}_1, \beta_1, \cdots, \boldsymbol{\alpha}_K, \beta_K\}$ corresponding to polynomial sub-functions $y_1 = p_1^{d_1}(x|\boldsymbol{\alpha}_1, \beta_1), y_2 = p_2^{d_2}(y_1|\boldsymbol{\alpha}_2, \beta_2), \cdots, y = p_K^{d_K}(y_{K-1}|\boldsymbol{\alpha}_K, \beta_K)$ by using the forward architecture, $x \mapsto y_1 \mapsto y_2 \cdots \mapsto y_{K-1} \mapsto y$. We adopt gradient-free differentiable evolution (DE) Rauf et al. (2021) to learn $\boldsymbol{\alpha}$ and $\beta$. DE uses the difference between individuals for mutation. Given the context vector $\boldsymbol{\lambda}^*$, we optimize $\boldsymbol{\alpha}_k$ and $\beta_k, 1 \leq k \leq K$ alternatively as:

$$\boldsymbol{\alpha}_k^\star = \arg\min_{\boldsymbol{\alpha}_k} \mathcal{L}(\boldsymbol{\alpha}_k | \boldsymbol{\lambda}^*), \ \boldsymbol{\alpha}_k | \boldsymbol{\lambda}^* = (\boldsymbol{\alpha}_1^*, \beta_1^*, \cdots, \boldsymbol{\alpha}_k, \cdots, \boldsymbol{\alpha}_K^*, \beta_K^*) \tag{4}$$

$$\beta_k^\star = \arg\min_{\beta_k} \mathcal{L}(\beta_k | \boldsymbol{\lambda}^*) + \gamma \cdot \beta_k^2, \ \beta_k | \boldsymbol{\lambda}^* = (\boldsymbol{\alpha}_1^*, \beta_1^*, \cdots, \beta_k, \cdots, \boldsymbol{\alpha}_K^*, \beta_K^*) \tag{5}$$

where $\mathcal{L}(\cdot)$ is the $\ell_1$ distance between $p^d(x)$ and $0.5 \cdot \text{sgn}(x)$. $\boldsymbol{\alpha}_k^{\star}$ and $\beta_k^{\star}$ are then used to update $\boldsymbol{\lambda}^*$. We introduce a *regularization* term for optimizing the scale parameters, where $\gamma$ is the scaling decay. Scale parameters can prevent polynomials from growing exponentially during the initial iterations. The decay helps guide the parameters gradually toward a value of one and eventually select promising coefficients. Our R-CCDE algorithm is detailed in Appendix D.2.

## 3.4 PAT: POLYNOMIAL-AWARE TRAINING

Replacing ReLU with EvoReLU in pre-trained neural networks injects *minor* approximation errors, which leads to performance loss. Fine-tuning can mitigate this performance loss by allowing the learnable weights to adapt to the approximation error. However, backpropagation through EvoReLU leads to exploding gradients due to high-degree polynomials. Thanks to precise *forward* approximation of EvoReLU, we can use gradients from the original non-arithmetic ReLU function for *backpropagation*, specifically, during *forward* pass, EvoReLU injects slight errors, which are captured by objective functions like cross-entropy loss. During the *backward* pass, we bypass EvoReLU and use ReLU to compute gradients to update the weights of the linear trainable layers (e.g., convolution or fully connected). We refer to this procedure, which bears similarity to STE Bengio et al. (2013) and QAT Jacob et al. (2018), as polynomial-aware training (PAT). The following pseudocode illustrates this procedure for a simple example, $\text{EvoReLU}(x) = x(0.5 + (f_3 \circ f_2 \circ f_1)(x))$. In the forward function, we first scale the coefficients of $f_3$ by $B$ so that the output range of $y$ is $[0, B]$. In the backward function, we compute the gradient $\partial y / \partial \text{ReLU(x)}$ instead of $\partial y / \partial \text{EvoReLU(x)}$.

```
def EvoReLU_forward(x, B):
    f3 = f3 * B
    y = f3(f2(f1(x)))
    y = x(0.5 + y)
    return y
```

```
def EvoReLU_backward(x, grad):
    y = ReLU(x)
    grad_y = dy/dx
    grad = grad * grad_y
    return grad
```

## 4 EXPERIMENTS

**Setup:** We benchmark AutoFHE on CIFAR-10 and CIFAR-100 Krizhevsky et al. (2009). Both datasets have 50,000 training and 10,000 validation images at a resolution of $32 \times 32$. The validation images are treated as private data and used only for evaluating the final networks. We randomly select 5,120 images from the training split as a *minival* Tan & Le (2021) dataset to guide the search process. The Top-1 accuracy on the minival dataset optimizes equation 3. In addition, PAT uses the training split to fine-tune polynomial networks. Finally, as our final result, we report the Top-1 accuracy on the *encrypted validation dataset under RNS-CKKS*. To evaluate AutoFHE under RNS-CKKS, we adopt the publicly available code of FHE-MP-CNN and adapt it for inference with layerwise EvoReLU. During inference, we keep track of the ciphertext levels and call the bootstrapping operation when the level reaches zero, thanks to the optimal placement of bootstrapping operations found by AutoFHE. For a fair comparison between AutoFHE and the baseline FHE-MP-CNN, we use the pre-trained network weights provided by FHE-MP-CNN.

**Hyperparameters:** For MOCoEv, we use a population size of 50 and run it for 20 generations. We set the probability of polynomial replacement to 0.5, the probability of polynomial removal to 0.4, and the probability of polynomial insertion to 0.1. For R-CCDE, we set the search domain of $\boldsymbol{\alpha}$ to $[-5, 5]$ and that of $\beta$ to $[1, 5]$. We set the population size for optimizing $\beta$ to 20. For $\boldsymbol{\alpha}$, we set the population size equal to $10\times$ the number of variables. We set the scaling decay to $\gamma = 0.01$ and the number of iterations to 200. For PAT, we use a batch size of 512 and weight decay of $5 \times 10^{-3}$ and clip the gradients to 0.5. We use learning rates of $5 \times 10^{-4}$ for CIFAR-10 and $2 \times 10^{-4}$ for CIFAR-100. During MOCoEV search, we set the fine-tuning epoch to one. After the search is done, we fine-tune searched polynomial networks for ten epochs. On one NVIDIA RTX A6000 GPU, the search process for ResNet-20/32/44/56 on CIFAR-10 took 59 hours, 126 hours, 200 hours, 281 hours, respectively. The search for ResNet-32 on CIFAR-100 took 140 hours.

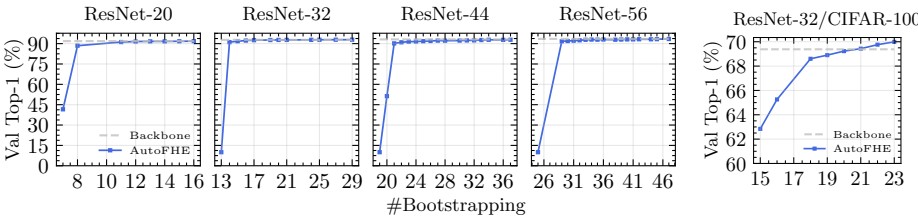

Figure 5: Pareto fronts of AutoFHE. We report the accuracy on plaintext validation datasets and the number of bootstrapping operations. Left: ResNet-20/32/44/56 on CIFAR-10; Right: ResNet-32 on CIFAR-100.

| Dataset | Backbone | | FHE-MP-CNN | | | | AutoFHE | | | |
|---------|----------|--------|------|-----------|-----------|-----------|------|-----------|-----------|-----------|
|         | Network  | Top-1  | Boot | Top-1*(%) | Inference | Amortized | Boot | Top-1(%)  | Inference | Amortized |
| CIFAR-10 | ResNet-20 | 91.86 | 18 | 91.31 | 3,532s | 71s | **13** | **91.39** | **2,643s** | **53s** |
|          | ResNet-32 | 92.80 | 30 | 92.40 | 5,768s | 115s | **20** | 92.25 | **4,201s** | **84s** |
|          |           |       |    |       |        |      | 21 | 92.68 | 4,435s | 89s |
|          |           |       |    |       |        |      | 29 | **92.88** | 5,386s | 108s |
|          | ResNet-44 | 93.13 | 42 | 92.65 | 7,732s | 155s | **38** | 92.04 | **7,209s** | **144s** |
|          | ResNet-56 | 93.49 | 54 | 93.07 | 9,837s | 197s | **47** | 93.27 | **8,684s** | **174s** |
| CIFAR-100 | ResNet-32 | 69.38 | 30 | **69.43** | 5,684s | 114s | **21** | 67.75 | **3,908s** | **78s** |
|           |           |       |    |       |        |      | 22 | 68.66 | 4,573s | 91s |
|           |           |       |    |       |        |      | 23 | 69.37 | 4,712s | 94s |

Table 2: AutoFHE under the RNS-CKKS scheme. Top-1* accuracy for FHE-MP-CNN, as reported in Lee et al. (2022a). The inference time for 50 images is evaluated on AMD EPYC 7H12 64-core processor using 50 threads. Boldface denotes the best criterion on a backbone network, like the best Top-1 accuracy and the least inference time; underline denotes AutoFHE outperforms FHE-MP-CNN.

## 4.1 PARETO FRONTS OF AUTOFHE

Figure 5 shows Pareto-effective solutions found by AutoFHE on CIFAR-10 and CIFAR-100 for different ResNet models. The trade-offs are between the Top-1 validation accuracy on plaintext data and the number of bootstrapping operations required for the corresponding homomorphic evaluation architecture. By optimizing the end-to-end network prediction function, AutoFHE adapts to the differing sensitivity of the activation layers to approximation errors and reduces the number of levels required compared to using the same high-degree AppReLU in all the layers. Thus, AutoFHE significantly reduces the number of bootstrapping operations. For ResNet-32 on CIFAR-10, AutoFHE removes 10 bootstrapping operations (33.33%) compared to FHE-MP-CNN with the negligible accuracy loss 0.08% compared to the original network with ReLUs. Lastly, AutoFHE provides a family of solutions offering different trade-offs rather than a single solution, thus providing flexible choices for practical deployments.

## 4.2 SECURE INFERENCE OF AUTOFHE UNDER RNS-CKKS

Due to the high computation cost of validating networks performance on encrypted data under the RNS-CKKS, we select nine solutions for evaluation on a machine with AMD EPYC 7H12 64-Core Processor and 1000 GB RAM. In Table 2, we evaluate three solutions for ResNet-32 on both CIFAR-10 and CIFAR-100 and evaluate one solution for ResNet-20/-44/56. We estimate the inference time for 50 images on 50 CPU threads. Amortized inference time is amortized runtime for each image. We report the Top-1 accuracy of AutoFHE on all (10,000) encrypted validation

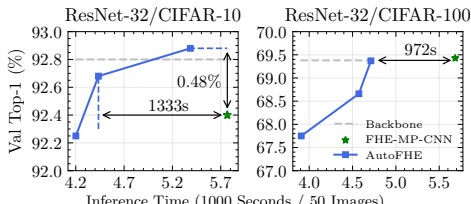

Figure 6: Pareto fronts of AutoFHE for ResNet-32 on CIFAR-10/-100 under RNS-CKKS.

images under the RNS-CKKS. We plot Pareto fronts on CIFAR-10 of AutoFHE versus the baseline in Figure 1. Figure 6 shows Pareto fronts of AutoFHE of ResNet-32 on encrypted CIFAR-10/-100. We observe that, on CIFAR-10, AutoFHE provides significant acceleration while having better accuracy or preserving accuracy. AutoFHE of ResNet-32 with 21 bootstrapping operations has slightly better accuracy than the baseline of ResNet-44 and accelerates inference by **3,297** seconds

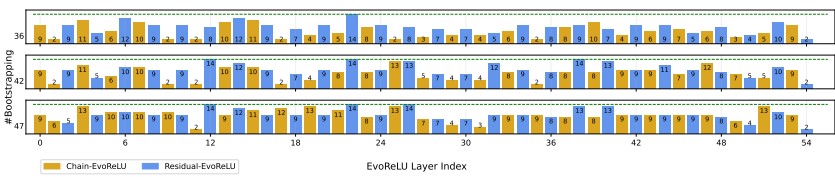

Figure 7: Depth consumption of layerwise EvoReLU for ResNet-56 with 36, 42 and 47 bootstrapping. The green dash line denotes the depth consumption of FHE-MP-CNN.

(**43**%). AutoFHE reduces inference time of ResNet-20, ResNet-32 (21 bootstrapping operations) and ResNet-56 by **25**%, **23**% and **12**% compared with the corresponding solutions of FHE-MP-CNN while improving accuracy up to **0.28**%. AutoFHE with 29 bootstrapping operations improves accuracy of ResNet-32 by a great margin **0.48**% while accelerating inference by 382 seconds (7%). AutoFHE can achieve a Top-1 accuracy of **91.39**% on encrypted CIFAR-10 under the RNS-CKKS at an amortized inference latency of under **one minute (53 seconds) per image**, which brings us closer towards practically realizing secure inference of deep CNNs under RNS-CKKS. On CIFAR-100, AutoFHE saves inference time by **972** seconds (**17**%) while preserving the accuracy. The experiments prove that AutoFHE can find Pareto-effective solutions that trade-off accuracy and inference time. Furthermore, the results validate our assumption that directly reducing the number of bootstrapping operations can effectively accelerate inference speed.

**Depth Distributions of Layerwise EvoReLU** for ResNet-56 with 36, 42, 47 bootstrapping operations are in Figure 7 (see Appendix H for more results). Chain EvoReLU and residual EvoReLU refer to EvoReLU in the chain connection and the residual connection shown in Figure3. We make three **observations**: **(1)** residual EvoReLUs consume more levels than chain EvoReLUs, suggesting that residual ReLU layers have less tolerance to approximation errors; **(2)** since pre-activations of chain EvoReLUs are normalized, they follow a tighter distribution and need smaller scaling values; **(3)** the last EvoReLU close to the output does not need high-precision approximation.

## 4.3 ABLATION STUDY

1. **Evaluating Co-evolution:** To evaluate the effectiveness of *co-evolution* in MOCoEv, we compare MOCoEv with a standard multi-objective evolution algorithm NSGA-II Deb et al. (2002) in Appendix E. The experimental results in Figure 8 and Table 4 show that co-evolution explores the optimization landscape of high-dimensional variables more effectively.

2. **Evaluating Layerwise Approximation:** To evaluate the effectiveness of adaptive *layerwise* approximation of AutoFHE, we compare it with uniformly distributed Minimax composite polynomials in Appendix F. Figure 9 shows that by exploiting the varying approximation sensitivity of different layers AutoFHE has a better trade-off than uniformly distributed Minimax polynomials.

3. **Evaluating AutoFHE on ImageNet:** We demonstrate the efficacy of AutoFHE on ImageNet Russakovsky et al. (2015), a large-scale high-resolution image dataset, in Appendix G. The experimental result in Figure 11 shows that AutoFHE can effectively trade-off the accuracy and the depth consumption on large-scale datasets with high-resolution images.

## 5 CONCLUSION

This paper introduced AutoFHE, an automated approach for accelerating CNNs on FHE and automatically designing a homomorphic evaluation architecture. AutoFHE seeks to approximate the end-to-end function represented by the network instead of approximating each activation function. We exploited the varying sensitivity of approximate activations across different layers in a network to jointly evolve composite polynomial activation functions and search for placement of bootstrapping operations for evaluation under RNS-CKKS. Experimental results over ResNets on CIFAR-10 and CIFAR-100 indicate that AutoFHE can reduce the inference time by up to 3,297 seconds (43%) while preserving the accuracy. AutoFHE also improves the accuracy by up to 0.48%. Although our focus in this paper was on ResNets, and consequently ReLU, AutoFHE is a general-purpose algorithm that is agnostic to the network architecture or the type of activation function.

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

## APPENDIX

In this appendix, we include the following:

- Appendix A: Notations;
- Appendix B: An expanded discussion of related work for secure inference;
- Appendix C: Background and related work for evolutionary algorithms;
- Appendix D: Implementation of the MOCoEv and R-CCDE algorithms in D.1 and D.2, respectively;
- Appendix E: Experimental details for evaluating co-evolution;
- Appendix F: Experimental details for evaluating layerwise comparison;
- Appendix G: Evaluation of AutoFHE on plaintext ImageNet;
- Appendix H: EvoReLUs of ResNet-56 on CIFAR-10.

## A  NOTATIONS

We list variable notations of the RNS-CKKS and EvoReLU in Table 3.

## B  RELATED WORK

**Secure Inference:** Secure inference is a promising solution for resolving the safety and privacy concerns in applications driven by deep learning as a service (DLaaS). Fully homomorphic encryption (FHE) and secure multiparty computation (MPC) are becoming the *de-facto* standard of secure inference of deep learning. Secure inference-based FHE Gilad-Bachrach et al. (2016); Brutzkus et al. (2019); Lou & Jiang (2021); Lee et al. (2022b;a) better takes advantage of the Cloud service provider's infrastructure. Customers only need to encrypt their private data, send ciphertexts to the Cloud, and decrypt the encrypted result. On the other hand, secure MPC Liu et al. (2017); Juvekar et al. (2018); Mishra et al. (2020); Lou et al. (2020); Ghodsi et al. (2021); Knott et al. (2021); Rathee et al. (2021) requires regular communication between customers and the Cloud. FHE cannot directly evaluate ReLU because it only allows arithmetic homomorphic addition and multiplication. However, secure MPC can evaluate ReLU using Garbled Circuits (GC) Yao (1986); Bellare et al. (2012) but suffers from high online computation and communication costs Mishra et al. (2020). *Adaption* of CNNs to secure inference by polynomial approximation of non-arithmetic functions, is a necessary pre-processing stage. Polynomial approximation enables us to homomorphically evaluate encrypted data on FHE, while it also can greatly reduce online computation and communication costs of secure MPC.

| Notation | Domain | Description |
|:---:|:---:|:---|
| $N$ | $\mathbb{Z}^+$ | degree of polynomial rings |
| $\ell$ | $\{0, 1, \cdots, L\}$ | level |
| $Q_\ell$ | $\mathbb{Z}^+$ | modulus $Q_\ell = \prod_{i=0}^{\ell} q_\ell$ where $q_\ell$ are primes |
| $\mathcal{R}_{Q_\ell}$ | | residual cyclotomic polynomial ring |
| $m$ | $\mathcal{R}_{Q_\ell}$ | plaintext |
| $e$ | $\mathcal{R}_{Q_\ell}$ | error |
| $\boldsymbol{c}$ | $\mathcal{R}_{Q_\ell}^2$ | ciphertext |
| $sk$ | $\mathcal{R}_{Q_\ell}^2$ | private key |
| $\langle \cdot, \cdot \rangle$ | | dot product |
| $[\cdot]_Q$ | | modular reduction function |
| $h$ | $\mathbb{Z}^+$ | Hamming weight |
| $x, y$ | $\mathbb{R}$ | scalar |
| $B$ | $\mathbb{R}$ | the maximum absolute value of pre-activations of ReLU |
| $p_\alpha(\cdot)$ | | a Minimax composite polynomial with precision $\alpha$ |
| $p^d(x)$ | | a composite polynomial $p^d(x) = (p_K^{d_K} \circ \cdots \circ p_k^{d_k} \circ \cdots \circ p_1^{d_1})(x)$ where $1 \leq k \leq K$ and degree $d = \prod_{k=1}^{K}$ |
| $p_k^{d_k}(x)$ | | a sub-polynomial function $p_k^{d_k}(x) = \beta_k \sum_{i=1}^{d_k} \alpha_i \mathrm{T}_i(x)$ $\alpha_i \in \mathbb{R}, \beta_i \in \mathbb{R}, T$ is the Chebyshev bases of the first kind |
| $\boldsymbol{d}$ | $\{d_i\}_{i=1}^{d_K}$ | degrees of all sub-polynomials |
| $\boldsymbol{\alpha}_k$ | $\{\alpha_i\}_{i=1}^{d_k}$ | coefficients of a sub-polynomial function |
| EvoReLU | | $y = \mathrm{EvoReLU}(x) = x \cdot \left(0.5 + p^d(x)\right), x \in [-1, 1], y \in [0, 1]$ |
| $\boldsymbol{\lambda}$ | | EvoReLU's parameters $\boldsymbol{\lambda} = (\boldsymbol{\alpha}_1, \beta_1, \cdots, \boldsymbol{\alpha}_K, \beta_K)$ |

Table 3: Variable Notations.

**Polynomial Approximation of ReLU:** A simple square activation function $x^2$ is used in CryptoNets Gilad-Bachrach et al. (2016), LoLa Brutzkus et al. (2019) and Deiphi Mishra et al. (2020). Faster CryptoNets Chou et al. (2018) exploits more accurate low-degree approximation $2^{-3}x^2 + 2^{-1}x + 2^{-2}$. SAFENet Lou et al. (2020) adopts $a_1x^3 + a_2x^2 + a_3x + a_4$ or $b_1x^2 + b_2x + b_3$ and uses SGD to train coefficients. When applying SGD to train low-degree polynomial coefficients and network weights simultaneously, polynomials easily lead to exploding gradients. On the other hand, low-degree polynomials need to train approximated networks from scratch, cannot use pre-trained weights, and has a big accuracy gap compared with ReLU networks. More recently, AESPA Park et al. (2022) proposes basis-wise normalization to address the problem of exploding gradients in low-degree polynomial approximated networks. Delphi and SAFENet apply population-based training (PBT) Jaderberg et al. (2017) to search for placement of polynomials. Because Delphi and SAFENet are evaluated under secure MPC, they maintain some ReLUs to preserve accuracy. SAFENet also observed that layerwise and channel-wise mixed-precision approximation could better take advantage of the varying sensitivity of different layers. Minimax composite polynomials Lee et al. (2021a;c) are specially designed to approximate ReLU under FHE with high precision using composite polynomials. FHE-MP-CNN Lee et al. (2022a) applies the Minimax composite polynomial with degree $\{15, 15, 27\}$ and proves it can maintain the performance of pretrained ResNets under the RNS-CKKS FHE scheme. Unlike the abovementioned methods, learn trainable weights, including coefficients of polynomial-only networks, by optimizing cross Entropy loss. Minimax is *function-level* approximation by optimizing the polynomial interpolation of ReLU. Given the depth, Minimax uses dynamic programming to optimize degrees of composite polynomials and applies improved multi-interval Remez algorithm Lee et al. (2021b) to solve coefficients. So, Minimax can achieve high approximation precision given depth. However, it neglects i) the learning ability of

neural networks to adapt to polynomial approximation, 2) the layerwise varying sensitivity, and 3) the combination of all polynomial activations in a network. In this paper, we consider both high-precision approximation and network performance. MOCoEv searches for degrees across all layers and directly optimizes validation accuracy. We consider function-level approximation using R-CCDE to minimize the $\ell_1$ distance. We use pre-trained ResNets and propose PAT to fine-tune network trainable weights to adapt to EvoReLUs for just a few epochs.

## C   EVOLUTIONARY SEARCH ALGORITHMS

**Evolutionary Algorithms (EAs)** are a class of search algorithms inspired by Darwin's natural selection. Each candidate solution is a *individual*. $NP$ individuals constitute the *population* with the population size $NP$. Individuals are assigned *fitness* related to the objective, like validation accuracy for image classification. Based on fitness, we randomly *select* mating individuals. *Crossover* combine mating individuals to generate *offspring*. *Offspring* can be further *mutated* to better exploit current knowledge. Finally, offspring is used to update the current population. We can iteratively repeat this process many times. Each iteration is called *generation*. The number of generations is a simple criterion for stopping the EA search.

**Multi-Objective EA (MOEA):** Given two $d$-dimensional vectors $\boldsymbol{x}_1$ and $\boldsymbol{x}_2$ for a minimization problem, if $\boldsymbol{x}_{1,i} \leq \boldsymbol{x}_{2,i}, \forall i \in \{1, 2, \cdots, d\}$ and $\boldsymbol{x}_{1,j} < \boldsymbol{x}_{2,j}, \exists j \in \{1, 2, \cdots, d\}$, $\boldsymbol{x}_1$ *dominates* $\boldsymbol{x}_2$ Srinivas & Deb (1994). It means $\boldsymbol{x}_1$ is better than $\boldsymbol{x}_2$. It is denoted as $\boldsymbol{x}_1 \prec \boldsymbol{x}_2$. Pareto front or Pareto-effective solutions are those not dominated by others. Delphi Mishra et al. (2020) and SAFENet Lou et al. (2020) combine two objectives (accuracy and ReLU replacement ratio) into a single objective by weighted sum. It is a widely-used trick to release multi-objective problems to a single objective. However, it only obtains a single solution balancing multiple objectives and cannot get Pareto-effective solutions. This is why we apply multi-objective search to obtain Pareto-effective solutions corresponding to different accuracy and latency requirements. EA is naturally well suited for multi-objective search due to population-based optimization. It allows us to obtain the entire set of Pareto-effective solutions in a single run. NSGA-II Deb et al. (2002) is the most well-known evolutionary multi-objective algorithm. The proposed MOCoEv adopts *nondominated sorting* and *crowding distance* from NSGA-II. The nondominated sorting can return all Pareto fronts, while crowding distance selects the uniformly distributed individuals within a Pareto front.

**Differentiable Evolution (DE)** is a gradient-free evolutionary algorithm used to optimize continuous variables Rauf et al. (2021). Given population $\boldsymbol{X} = \{\boldsymbol{x}_1, \boldsymbol{x}_2, \cdots, \boldsymbol{x}_{NP}\}$, where each individual $\boldsymbol{x}_\pi \in \mathbb{R}^d, 1 \leq \pi \leq NP$, the mutation, crossover and selection of DE are defined as:

$$\text{Mutation: } \boldsymbol{v} = \boldsymbol{x}_{\pi_1} + F \cdot (\boldsymbol{x}_{\pi_2} - \boldsymbol{x}_{\pi_3}), 1 \leq \pi_1, \pi_2, \pi_3 \leq NP$$

$$\text{Crossover: } \boldsymbol{u}_j = \begin{cases} \boldsymbol{v}_j, & \mathcal{U}(0,1) \leq CR \\ \boldsymbol{x}_{\pi_1,j}, & \text{Otherwise} \end{cases}, \quad 1 \leq j \leq d \tag{6}$$

$$\text{Selection: } \boldsymbol{u} = \begin{cases} \boldsymbol{u}, & \mathcal{F}(\boldsymbol{u}) \geq \mathcal{F}(\boldsymbol{x}_{\pi_1}) \\ \boldsymbol{x}_{\pi_1}, & \text{Otherwise} \end{cases}$$

where $F$ is scaling factor, $CR$ is crossover rate, $\mathcal{F}(\cdot)$ is the fitness evaluation function, and $\mathcal{U}(0,1)$ is the uniform distribution between 0 and 1.

**Cooperative Co-Evolution (CC)** algorithms were proposed to address the challenge of optimizing high-dimensional variables Yang et al. (2008); Mei et al. (2016); Ma et al. (2018). Co-evolution decomposes the high-dimensional optimization problem into low-dimensional sub-problems. Then, we can apply EAs or DE to solve sub-problems for discrete or continuous variables, respectively. CC includes two major stages, *decomposition* and *cooperative evaluation*. Decomposition refers to grouping variables. Simple grouping strategies include random or interaction-based (gradient-based) grouping Mei et al. (2016). When the proposed R-CCDE searches for parameters $\boldsymbol{\lambda} = (\boldsymbol{\alpha}_1, \beta_1, \cdots, \boldsymbol{\alpha}_K, \beta_K)$ of EvoReLU$(x, \boldsymbol{\lambda}|\boldsymbol{d})$, we decompose $\boldsymbol{\lambda}$ to $\boldsymbol{\alpha}_1, \beta_1, \cdots, \boldsymbol{\alpha}_K, \beta_K$ corresponding to polynomial sub-functions $y_1 = p_1^{d_1}(x|\boldsymbol{\alpha}_1, \beta_1), y_2 = p_2^{d_2}(y_1|\boldsymbol{\alpha}_2, \beta_2), \cdots, y = p_K^{d_K}(y_{K-1}|\boldsymbol{\alpha}_K, \beta_K)$. Because the scaling parameter $\beta$ is used to adjust the amplitude of polynomials, we evolve $\beta$ followed by $\boldsymbol{\alpha}$. We maintain $2K$ populations for $\boldsymbol{\alpha}_1, \beta_1, \cdots, \boldsymbol{\alpha}_K, \beta_K$, separately. These populations are called *sub-populations* or *species* in CC. This decomposition of R-CCDE takes advantage of the forward architecture of composited polynomials, $x \mapsto y_1 \mapsto y_2 \cdots \mapsto y_{K-1} \mapsto y$.

When the proposed MOCoEV searches for $\boldsymbol{D} = \{\boldsymbol{d}_1, \boldsymbol{d}_2, \cdots, \boldsymbol{d}_L\}$ to minimize the objective $\{1 - \mathrm{Acc}_{val}(f(\boldsymbol{\omega}^*); \boldsymbol{\Lambda}^*(\boldsymbol{D}), \boldsymbol{D}), \mathrm{Boot}(\boldsymbol{D})\}$, we evolve sub-populations for $\boldsymbol{d}_1, \boldsymbol{d}_2, \cdots, \boldsymbol{d}_L$, separately. It decomposes original problems with dimension $114 \sim 330$ to dimension six and greatly reduces the search space size from $10^{79} \sim 10^{230}$ to $10^4$. Cooperative evaluation refers to cooperatively evaluating an individual's fitness in a sub-population. We should take into account other sub-populations when evaluating the individual. R-CCDE is a single objective optimization, so we maintain a *context vector* Mei et al. (2016) $\boldsymbol{\lambda}^* = (\boldsymbol{\alpha}_1^*, \beta_1^*, \cdots, \boldsymbol{\alpha}_K^*, \beta_K^*)$. When evaluate $\boldsymbol{\alpha}_k$ or $\beta_k$, we just need to replace the corresponding $\boldsymbol{\alpha}_k^*$ or $\beta^*$ and assign $\boldsymbol{\alpha}_k$ or $\beta$ with the fitness of $(\boldsymbol{\alpha}_1^*, \beta_1^*, \cdots \boldsymbol{\alpha}_k \cdots, \boldsymbol{\alpha}_K^*, \beta_K^*)$ or $(\boldsymbol{\alpha}_1^*, \beta_1^*, \cdots \beta_k \cdots, \boldsymbol{\alpha}_K^*, \beta_K^*)$. In the beginning, the context vector is randomly initialized. Finally, R-CCDE outputs the context vector as the searched result. We extend the context vector for multi-objective optimization. MOCoEv maintains context vectors that are the current Pareto Front. Therefore, MOCoEv can effectively improve the Pareto Front using CC.

# D  THE PROPOSED SEARCH ALGORITHMS

## D.1  MULTI-OBJECTIVE COOPERATIVE EVOLUTION

Algorithm 1 shows the details of our proposed MOCoEv search algorithm. MOCoEv takes as input a neural network $f$ with $L$ ReLUs that will be replaced by EvoReLUs, the number of sub-functions of a composite polynomial $K$, the population size $NP$, the number of iterations $T$, the initial population size $N_0$. $N_0 \gg N$ because random initialization will generate *invalid* individuals. Invalid individuals refer to those that will lead to negative levels. The dataset, like CIFAR-10, with training and validation datasets, will be used. We will randomly sample a subset from the training dataset as *minival* dataset used for search. The training dataset will be used for fine-tuning. During search and fine-tuning, the validation dataset is strictly unseen. We will report the Top-1 validation accuracy on the validation dataset as the final result. MOCoEv will output the Pareto front, namely the population. The population is composed of non-dominated individuals with varying numbers of bootstrapping.

---

**Algorithm 1:** MOCoEv

---

**input**  : The Network $f$ with $L$ ReLUs, the number of sub-functions of a composite polynomial $K$, the population size $NP$, the number of iterations $T$, the initial population size $N_0 \gg NP$, the replace probability $P_{\mathrm{replace}}$, the remove probability $P_{\mathrm{remove}}$, the insert probability $P_{\mathrm{insert}}$, the training dataset Training, the mini-validation dataset Minival;

**output** : The Pareto front Population;

**initial**  : Population $\{\boldsymbol{D}_1, \boldsymbol{D}_2, \cdots, \boldsymbol{D}_{N_0}\} \leftarrow$ LHS$(N_0, L, K)$ where $\boldsymbol{D} = \{\boldsymbol{d}_1, \cdots, \boldsymbol{d}_L\}$;
    **foreach** $d$ *in* $\boldsymbol{D}$ **do** $\lambda \leftarrow$ R-CCDE$(\boldsymbol{d})$ ;
    **foreach** $\boldsymbol{D}$ *in* Population **do** Acc $\leftarrow$ Evalaute$(f(\boldsymbol{D}, \boldsymbol{\Lambda}),$ Minival$)$ ;
    Population $\leftarrow$ Pareto(Population, Acc, $NP$)
    **foreach** $\boldsymbol{D}$ *in* Population **do** $\omega \leftarrow$ PAT$(f(\boldsymbol{D}, \boldsymbol{\Lambda}),$ Training$)$ ;
    **foreach** $\boldsymbol{D}$ *in* Population **do** Acc $\leftarrow$ Evalaute$(f(\boldsymbol{\omega}, \boldsymbol{D}, \boldsymbol{\Lambda}),$ Minival$)$ ;

**for** $t \leftarrow 1$ **to** $T$ **do**
    **for** $i \leftarrow 1$ **to** $L$ **do**
        Offspring $\leftarrow$ Select(Population) ;
        Offspring $\leftarrow$ Crossover(Offspring) ;
        Offspring $\leftarrow$ Mutate(Offspring$[:, i]$) ;
        **foreach** $d$ *in* Offspring $[:, i]$ **do** $\lambda \leftarrow$ R-CCDE$(\boldsymbol{d})$ ;
        **foreach** $\boldsymbol{D}$ *in* Offspring **do** $\omega \leftarrow$ PAT$(f(\boldsymbol{D}, \boldsymbol{\Lambda}),$ Training$)$ ;
        **foreach** $\boldsymbol{D}$ *in* Offspring **do** Acc $\leftarrow$ Evalaute$(f(\boldsymbol{\omega}, \boldsymbol{D}, \boldsymbol{\Lambda}),$ Minival$)$ ;
        Population $\leftarrow$ Pareto(Population +Offspring, Acc, $NP$) ;

---

In the initialization, we randomly initialize the population with $N_0$ individuals, $\{\boldsymbol{D}_1, \boldsymbol{D}_2, \cdots, \boldsymbol{D}_{N_0}\}$ where $\boldsymbol{D}_j = \{\boldsymbol{d}_1, \cdots, \boldsymbol{d}_L\} \in \mathbb{Z}^{L \times K}, 1 \leq j \leq N_0$. $\boldsymbol{d}_i, 1 \leq i \leq L$ is the degrees of a composite polynomial and is randomly sampled using the Latin hypercube sampling (LHS) method. The composite polynomials with the layer index $i$ constitute the $i$-th sub-populations of CC. The proposed R-CCDE searched for coefficients of composite polynomials. The Pareto function is first to use nondominated sorting to find Pareto fronts and then use crowding distance to select individuals given

the population size $NP$. In iteration $t$, we sequentially evolve EvoReLUs one by one. Given $i$-th EvoReLU, we first randomly select mating individuals from the population based on their accuracy on the minival dataset. We crossover mating individuals to generate offspring. Crossover operates at the network level. Then, we mutate the $i$-th sub-population. We randomly replace, remove and insert polynomials of the $i$-th sub-population with the probability $P_{\text{replace}}$, $P_{\text{remove}}$ and $P_{\text{insert}}$, respectively. So, we need to apply R-CCDE to search coefficients of the $i$-th sub-population and also use PAT to fine-tune the network. We evaluate the fine-tuned networks on the minival dataset. Finally, we use $\text{Pareto}$ function to obtain the new Pareto front given the population size $NP$. Individuals in the Pareto front will be used to replace the population.

## D.2 REGULARIZED CO-OPERATIVE DIFFERENTIAL CO-EVOLUTION

---

**Algorithm 2:** R-CCDE

---

**input** : A composite polynomial $p^d(x) = (p_K^{d_K} \circ p_{K-1}^{d_{K-1}} \circ \cdots \circ p_1^{d_1})(x)$ with parameters
$\boldsymbol{\lambda} = \{\boldsymbol{\alpha}_1, \beta_1, \cdots, \boldsymbol{\alpha}_K, \beta_K\}$, the target non-arithmetic function $q(x)$, the number of iterations $T$, the scaling decay $\gamma$;
**output** : The context vector $\boldsymbol{\lambda}^*$;
**initial** : $\boldsymbol{\lambda}^* \leftarrow \texttt{LHS}$

**for** $t \leftarrow 1$ **to** $T$ **do**
    **for** $k \leftarrow 1$ **to** $K$ **do**
        $\boldsymbol{\alpha}_k^\star = \arg\min_{\boldsymbol{\alpha}_k} \mathcal{L}_{p^d,q}(\boldsymbol{\alpha}_k|\boldsymbol{\lambda}^*)$, $\boldsymbol{\alpha}_k|\boldsymbol{\lambda}^* = (\boldsymbol{\alpha}_1^*, \beta_1^*, \cdots \boldsymbol{\alpha}_k, \cdots, \boldsymbol{\alpha}_K^*, \beta_K^*)$;
        $\boldsymbol{\lambda}^* \leftarrow (\boldsymbol{\alpha}_1^*, \beta_1^*, \cdots \boldsymbol{\alpha}_k^\star, \cdots, \boldsymbol{\alpha}_K^*, \beta_K^*)$;
        $\beta_k^\star = \arg\min_{\beta_k} \mathcal{L}_{p^d,q}(\beta_k|\boldsymbol{\lambda}^*) + \gamma \cdot \beta_k^2$, $\beta_k|\boldsymbol{\lambda}^* = (\boldsymbol{\alpha}_1^*, \beta_1^*, \cdots, \beta_k, \cdots, \boldsymbol{\alpha}_K^*, \beta_K^*)$;
        $\boldsymbol{\lambda}^* \leftarrow (\boldsymbol{\alpha}_1^*, \beta_1^*, \cdots \beta_k^\star, \cdots, \boldsymbol{\alpha}_K^*, \beta_K^*)$;

---

Algorithm 2 details the proposed R-CCDE searches for coefficients of a composite polynomial. R-CCDE takes as input a composite polynomial $p^d(x)$, the target non-arithmetic function $q(x)$, the number of iterations $T$ and scaling decay parameter $\gamma$. In this paper, $q(x) = 0.5 \cdot \text{sgn}(x)$. R-CCDE will output the context vector $\boldsymbol{\lambda}^*$ as result. The composite polynomial $p^d(x) = (p_K^{d_K} \circ p_{K-1}^{d_{K-1}} \circ \cdots \circ p_1^{d_1})(x)$ has learnable parameters $\boldsymbol{\lambda} = \{\boldsymbol{\alpha}_1, \beta_1, \cdots, \boldsymbol{\alpha}_K, \beta_K\}$. $\boldsymbol{\alpha}_k = \{\alpha_1, \cdots, \alpha_{d_k}\}$ and $\beta_k, 1 \leq k \leq K$ satisfy $p_k^{d_k}(x) = \beta_k \sum_{i=1}^{d_k} \alpha_i \text{T}_i(x)$. We apply LHS to initialize sub-populations of each $\boldsymbol{\alpha}_k$ and $\beta_k$. The population size of $\boldsymbol{\alpha}_k$ sub-populations is equal to $10 \times \lfloor d_k + 1 \rfloor / 2$, where $\beta_k$ is 20. We set $\beta_K = 1$ and not learnable. Given iteration $t$ and sub-population index $k$, we apply DE to solve $\arg\min_{\boldsymbol{\alpha}_k} \mathcal{L}_{p^d,q}(\boldsymbol{\alpha}_k|\boldsymbol{\lambda}^*)$ where $\mathcal{L}_{p^d,q}$ is the $\ell_1$ distance between $p^d(x)$ and $q(x)$. $\boldsymbol{\alpha}_k|\boldsymbol{\lambda}^*$ means to use the current context vector $\boldsymbol{\lambda}^*$ but only search for the corresponding $\boldsymbol{\alpha}_k$. The solution $\boldsymbol{\alpha}_k^\star$ evolved by DE is used to update the context vector $\boldsymbol{\lambda}^* = (\boldsymbol{\alpha}_1^*, \beta_1^*, \cdots \boldsymbol{\alpha}_k^\star, \cdots, \boldsymbol{\alpha}_K^*, \beta_K^*)$. Similarly, we also evolve $\beta_k$. After obtaining $\boldsymbol{\lambda}^*$, we can scale $\boldsymbol{\alpha}_k$ by $\beta_k$ and obtain the coefficients of the composite polynomial in terms of the first kind of Chebyshev basis.

## E COMPARISON OF SEARCH ALGORITHMS

To solve the high-dimensional search problem $\min_{\boldsymbol{D}} \{1 - \text{Acc}_{val}(f(\boldsymbol{\omega}^*); \boldsymbol{\Lambda}^*(\boldsymbol{D}), \boldsymbol{D}), \text{Boot}(\boldsymbol{D})\}$, we propose MoCoEv by using cooperative co-evolution to decompose high-dimensional optimization problem to low-dimensional sub-problems. Because we adopt non-dominated sorting and crowding distance from NSGA-II Deb et al. (2002) to obtain Pareto-effective solutions. NSGA-II is a fair comparison to demonstrate the efficacy of MoCoEv. We conduct the search experiments of ResNet-20 and ResNet-32 on plaintext CIFAR-10. When we use NSGA-II, we set the same hyper-parameters except for increasing the population size by the number of ReLUs. However, we cannot control the number of polynomials being evaluated, the number of evaluations on the minival dataset, and the number of fine-tuning on the training dataset. So, we use the wall-clock search time to make computation comparable, as shown in Table 4.

The upper row of Figure 8 shows the Pareto fonts of NSGA-II and AutoFHE. In terms of the trade-off of the Top-1 validation accuracy ($> 80\%$) versus the number of bootstrapping operations, NSGA-II

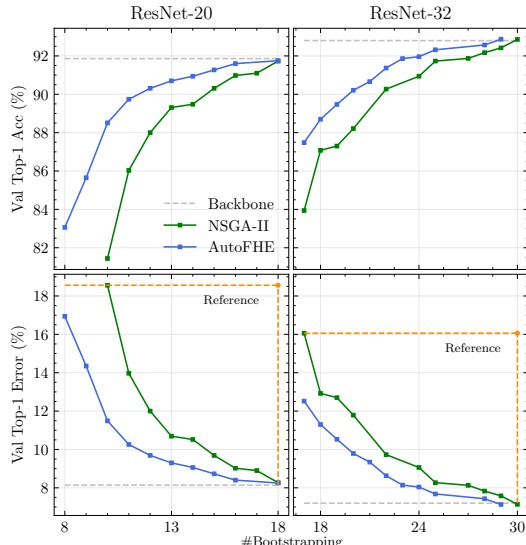

Figure 8: Comparison between AutoFHE with Co-evolution and NSGA-II.

| Backbone | | NSGA-II Deb et al. (2002) | | | | | AutoFHE | | | | | |
|---|---|---|---|---|---|---|---|---|---|---|---|---|
| Network | Top1 | HV | Time | #Iter | #Poly | #Eval | #Tune | HV | Time | #Iter | #Poly | #Eval | #Tune |
| ResNet-20 | 91.86 | 55.13 | 2 days 17 hrs | 12 | 219k | 9k | 8k | 72.98 | 2 days 17 hrs | 20 | 24k | 18k | 18k |
| ResNet-32 | 92.28 | 74.92 | 6 days 14 hrs | 15 | 713k | 14k | 14k | 92.89 | 6 days 5 hrs | 20 | 36k | 30k | 30k |

Table 4: The ablation experiment of search algorithms on plaintext CIFAR-10.

is inferior to AutoFHE. It proves the effectiveness of *co-evolution*. Hypervolume (HV) Fonseca et al. (2006) is used to compare NSGA-II and AutoFHEquantitatively. Hypervolume denotes the volume dominated by the Pareto front. The bigger is HV, the better the Pareto Front. The bottom row of Figure 8 shows the trade-off between the Top-1 validation error and the number of bootstrapping. We compute the HV with respect to the reference points, $(18.00, 18.56)$ for ResNet-20 and $(30.00, 16.06)$ for ResNet-32, respectively. Table 4 shows AutoFHE has better HV values than NSGA-II. These ablation experiments prove that co-evolution facilitates the high-dimensional multi-objective search.

## F    EVALUATION OF LAYERWISE RELU APPROXIMATION

To demonstrate the efficacy of *layerwise* approximation of EvoReLUs, we compare AutoFHE with the uniformly approximated networks. We adopt the Minimax composite polynomials Lee et al. (2021a;c) from precision 4 to 14 and use them to replace ReLUs uniformly. However, the Minimax polynomials require a re-design of homomorphic evaluation architecture for all composite polynomials with different precision. For example, FHE-MP-CNN uses the polynomial with precision 13 and designs the suitable homomorphic evaluation architecture. To fairly compare the layerwise EvoReLUand the uniformly distributed Minimax polynomial, we use the number of depths consumed by polynomials rather than the number of bootstrapping as the criterion. We report the Top-1 validation accuracy on CIFAR-10 as the estimated performance under the RNS-CKKS. Hence, we used the search objective $\min_{\boldsymbol{D}}\{1 - \text{Acc}_{val}(f(\boldsymbol{\omega^*}); \boldsymbol{\Lambda^*}(\boldsymbol{D}), \boldsymbol{D}), \text{Depth}(\boldsymbol{D})$, where $\text{Depth}(\cdot)$ is the total number of depth consumed by polynomials. On the other hand, in this ablation, we do NOT use PAT to fine-tune networks to make the comparison fair.

The upper row of Figure 9 shows Pareto fronts of Minimax and AutoFHE in terms of Top-1 validation accuracy and the number of depth. From the bottom row of Figure 9, we compute the hypervolume values with respect to the reference point $(285, 90)$ for ResNet-20 and the reference point $(372, 90)$ for ResNet-32, respectively. AutoFHE has HV $1.58 \times 10^4$ better than Minimax HV $1.06 \times 10^4$ on ResNet-20, while AutoFHE HV is $1.84 \times 10^4$ compared with Minimax HV $1.00 \times 10^4$. These experimental results prove: 1) the layerwise approximation is better than uniform approximation; 2)

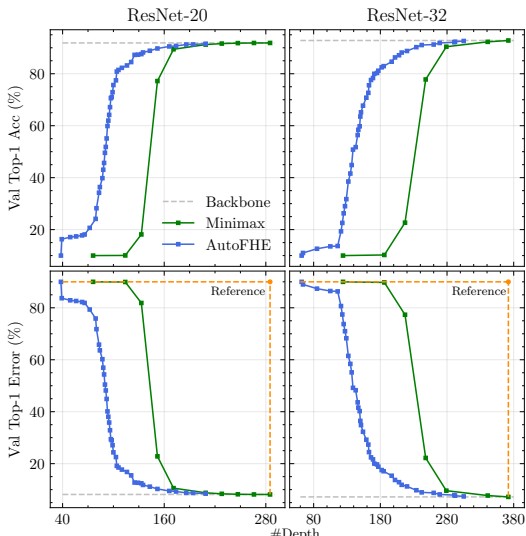

Figure 9: Evaluation of AutoFHE and layerwise Minimax.

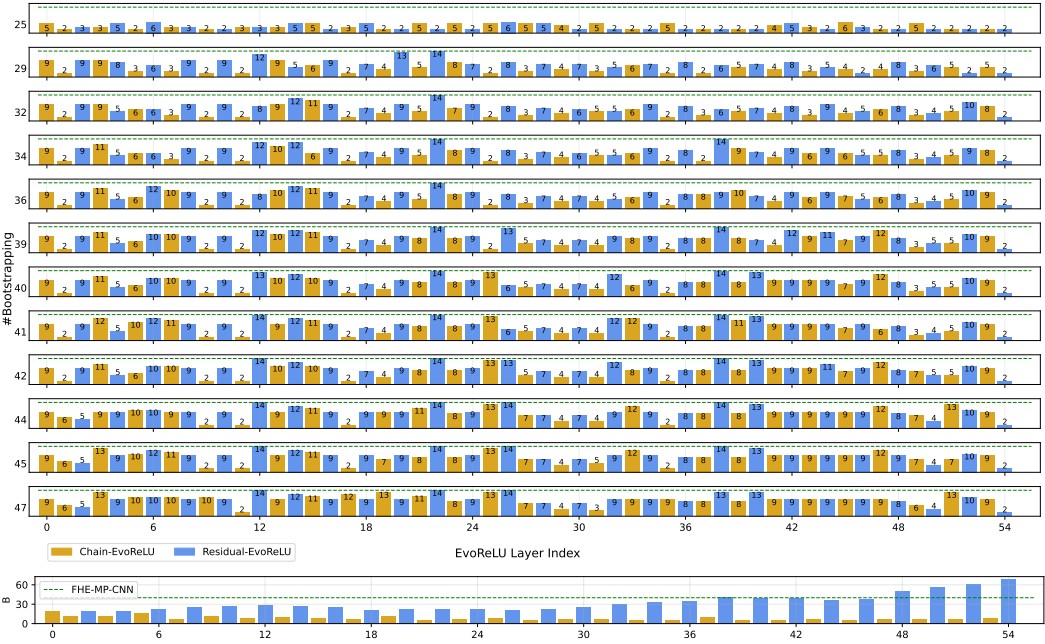

Figure 10: Depth consumption distribution of EvoReLUs of ResNet-56. Upper: depth consumption distributions of layerwise EvoReLUs of different bootstrapping consumption. Bottom: the distribution of scaling parameters (B) of layerwise EvoReLUs. The green dashed lines show the depth consumption or B of AppReLUs of FHE-MP-CNN.

the approximation of AutoFHE is also precise and AutoFHE's performance is not simply because of fine-tuning.

## G EVALUATE AUTOFHE ON PLAINTEXT IMAGENET

It is not practical to evaluate high-resolution images under the RNS-CKKS scheme due to the extremely high memory footprint and computational complexity constraints. To realize the goal of practically processing high-resolution images in the encrypted domains, we need advancements in

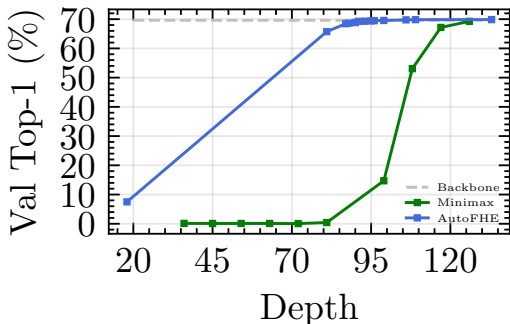

Figure 11: Evaluate AutoFHE over ResNet-18 on plaintext ImageNet.

RNS-CKKS primitives, custom hardware for FHE, more efficient packing algorithms, etc. This is the reason why all current works use CIFAR to benchmark performance. We evaluate AutoFHE on plaintext ImageNe Russakovsky et al. (2015) to demonstrate its efficacy on the large-scale high-resolution dataset. We use the ResNet-18 model provided by Pytorch. It has 9 ReLUs and 1 MaxPooling. We replace the non-arithmetic MaxPooling by the arithmetic AvgPooling and train it from scratch for 90 epochs. Its Top-1 accuracy is 69.62% slightly lower than its MaxPooling version, 69.76%. We set the number of generations to 10, the size of the minival dataset to 2,560, and the population size to 30. Other hyper-parameters are as same as CIFAR experiments. We turn off fine-tuning during the search and fine-tune the final result for one epoch. The search experiment took 18 hours. We estimate accuracy on the plaintext validation dataset and use depth consumption as the inference cost under the RNS-CKKS. We adopt the Minimax polynomials with precision from 4 to 13 Lee et al. (2021c) as our baseline and set $B = 100$. Figure 11 shows Pareto fronts of AutoFHE and Minimax. AutoFHE has 69.26% accuracy with a depth consumption of 91, while Minimax consumes 126 depth to have the same accuracy. AutoFHE can reduce 28% depth consumption. When depth consumption is equal to 99, Minimax has an accuracy 14.71%. AutoFHE reports accuracy 65.56% with the same depth consumption. This experiment demonstrates that AutoFHE can effectively trade off the accuracy and the depth consumption on large-scale high-resolution datasets.

## H    EVORELUS OF RESNET-56

Figure 10 shows distributions of depth consumption of EvoReLUs of Pareto-effective solutions with varying numbers of bootstrapping operations. From the upper panel of Figure 10, MoCoEv exploits the layerwise variable approximation sensitivity and assigns different depths to each EvoReLU. So, AutoFHE can reduce depth consumption, save the number of bootstrapping operations and further accelerate inference. From the bottom panel of Figure 10, it shows different distributions of pre-activations. It proves we can use smaller B values and lower-degree polynomials to have the same precision, namely $B' \cdot 2^{-\alpha'} = B \cdot 2^{-\alpha}$, where $B' < B$ and $\alpha' < \alpha$. The pre-activations of residual EvoReLU are not normalized, so its B values are bigger than chain EvoReLU. So, residual EvoReLUs prefer higher-degree polynomial approximation with more depth consumption to maintain approximation precision.

Figure 12 and Figure 13 show EvoReLUs of different layers of ResNet-56. We include Pareto-effective solutions with the number of bootstrapping operations from 26 to 52. From Figure 12 and Figure 13, high-precision solutions consume more depth and approximate ReLUs precisely. Low-precision solutions use low-degree polynomials to reduce depth consumption. From EvoReLU approximation, we learn how AutoFHE can trade off accuracy and inference speed.

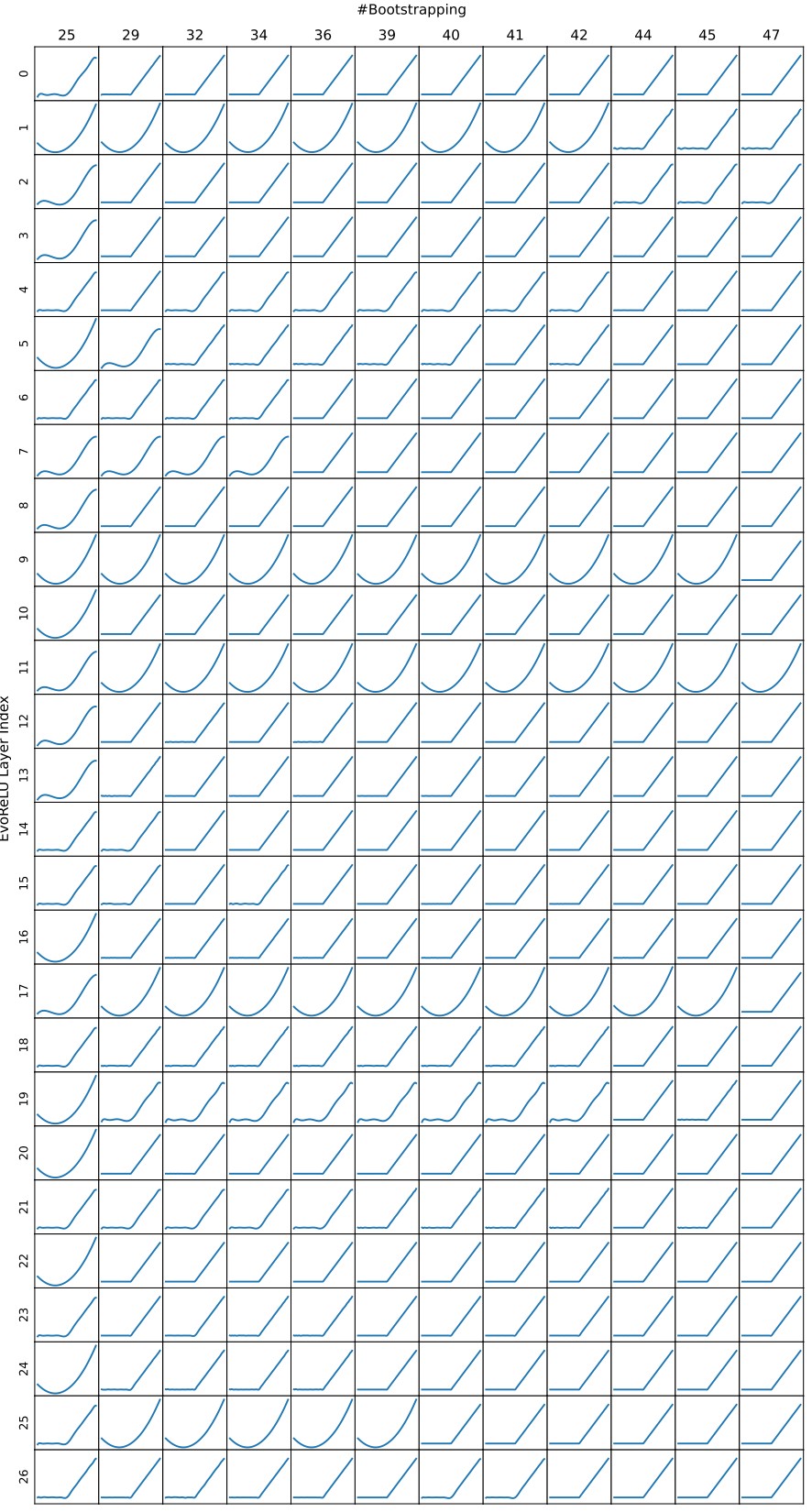

Figure 12: EvoReLUs of ResNet-56 from layer 0 to 26.

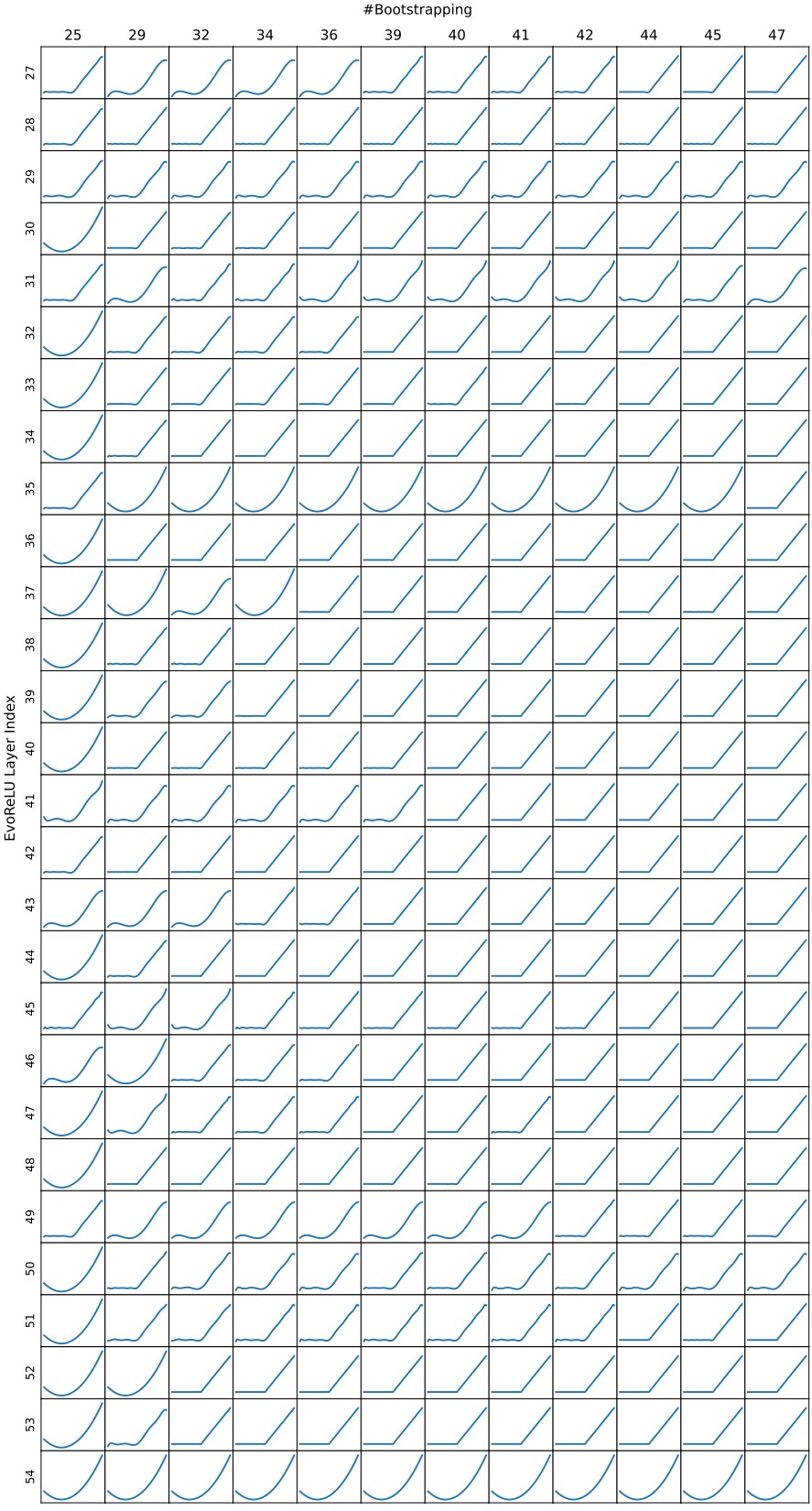

Figure 13: EvoReLUs of ResNet-56 from layer 27 to 54.

