# OpenReview forum: "AutoFHE: Automated Adaption of CNNs for Efficient Evaluation over FHE"
_ICLR.cc/2023/Conference — Submitted to ICLR 2023_

### Official Review · Reviewer_b7YQ · 2022-10-22

**Confidence:** 4
**Correctness:** 3
**Technical Novelty And Significance:** 2
**Empirical Novelty And Significance:** 2
**Recommendation:** 5

**Clarity, Quality, Novelty And Reproducibility:**

**Clarity**: Average, as the intro misses another domain of research to solve similar problem. However, the preliminaries are well written.

**Quality**: Good in terms of problem identification, average in terms of providing intriguing solution.

**Novelty**: Average.

**Reproducibility**: Code not shared. Algorithm is provided in Appendix.

**Details Of Ethics Concerns:**

N/A.

**Strength And Weaknesses:**

### Strengths

1. In the current era of data and model IP protection, the problem statement of minimizing latency for secure and private inference is gullible and the paper is decently written.
2. The issue with FHE-MP-CNN is well identified, particularly, the point of same high order approx. for all ReLUs.
3. The multi objective problem formulation is well inspired.

### Weakness

1. Abstract introduced too much technicality as compared to its size with many technical terms. This definitely raises more question as opposed to providing a short overview of the paper. Some terms are not even clear, e.g. what's the full form of RNS-CKSS (in context of abstract).

2. There has been few research in the domain of ReLU reduction in the context of private inference, including selective network linearization (SNL) [1], DeepReDuce [2] etc. These line of research definitely aligns with the current problem statement, hence, it would be interesting to see how efficient approximation of ReLU and removal of ReLUs go hand in hand in the context of FHE.

3. If EvoReLU does not help in reducing latency, then is it replaced (from AppReLU) to improve accuracy? Or there is something I am missing? Does EvoReLU introduces polynomial error based on layer sensitivity? Please clarify, as the authors had mentione dApprox ReLU uses similar approx error for all ReLU layers.

4. If you replace the EvoReLU with ReLU during backward pass, wont it introduce significant error? How that error is compensated in the framework? Why EvoReLU introduce exploding gradient issue, and why it can't be solved by traditional mean of solving exploding gradients? STE was introduced to primarily deal with non differentiable functions, not to handle exploding gradients. So, the citation of STE in this context is confusing!

5. MoCoEv and Sec. 3.3 are written in a hurried way without properly mentioning the inspiration of the proposed solution. I find no discussion on possible alternatives and why we should buy this "REGULARIZED COOPERATIVE DIFFERENTIABLE CO-EVOLUTION" as the best potential solution.

6. There is no pareto frontier of the search space, either for bootstrap or the EvoReLU polynomial choices. There is no latency vs accuracy plot either.

7. Details of latency improvement computation is missing (the one mentioned as 17% and 22%).

8. Additional results on higher resolution datasets are missing. Both CIFAR-10 and 100 are of size 32x32x3.

9. The authors should also compare with more advanced search methods, instead of only comparing with NSGA II, which is not necessarily the state-of-the-art.

### Other small issues

Please clarify (in the preliminaries) why bootstrapping operation is costly in a bit more details.


[1] selective network linearization, ICML 2022.

[1]  DeepReDuce, ICML 2021.

**Summary Of The Paper:**

The paper proposes an automatic search based approach to design latency efficient CNN models for FHE. In particular, the paper presents a three stage approach of the search and fine tune: i) a multi-objective co-evolutionary (MOCoEv) search algorithm to maximize validation accuracy and minimize the number of bootstrapping operations, ii) a gradient-free search algorithm, R-CCDE, to optimize EvoReLU
coefficients, and iii) polynomial-aware training (PAT) to fine-tune polynomial-only CNNs for one epoch to adapt trainable weights to EvoReLUs. Demonstration on CIFAR-10 and CIFAR-100 shows the efficacy of the proposed method.

**Summary Of The Review:**

The paper targets to solve the latency bottleneck issue of bootstrap and ReLU units forFHE based private inference. The problem is well motivated. However, the paper lacks ease of flow and clarity in writing. Details of error analysis for approx EvoReLU, latency evaluations, additional results, reason behind choice of search optimization are missing. Thus, despite good promise, the paper falls significantly below acceptance threshold.

---

> ### Author Response · Authors · 2022-11-16
> **Response to Reviewer b7YQ (Part I)**
>
> ### We appreciate your efforts and constructive comments. We followed your suggestions to improve our paper and addressed your concerns.
>
> > **Q1:** Abstract introduced too much technicality as compared to its size with many technical terms. This definitely raises more question as opposed to providing a short overview of the paper. Some terms are not even clear, e.g. what's the full form of RNS-CKSS (in context of abstract).
>
> **A1:** We explained RNS-CKKS in an updated version. We understand that some readers may not have a cryptographic background, so we explained all the necessary FHE background in the preliminary section.
>
> > **Q2:** There has been few research in the domain of ReLU reduction in the context of private inference, including selective network linearization (SNL), DeepReDuce etc. These line of research definitely aligns with the current problem statement, hence, it would be interesting to see how efficient approximation of ReLU and removal of ReLUs go hand in hand in the context of FHE.
>
> **A2:** Thank you for sharing these two pieces of literature. We also mentioned similar works, Delphi [1] and SAFENet [2] (see Appendix B). SNL, DeepReDuce, and SAFENet are based on Delphi’s framework. These works and our AutoFHE differ in two crucial aspects: **(1) Different secure inference schemes.** Delphi is based on secure MPC, while AutoFHE focuses on FHE. Secure MPC can evaluate ReLU using Garbled Circuits, but FHE cannot evaluate ReLU. **(2) Different motivation and research goals.** SNL, DeepReDuce, SAFENet, and Delphi **still maintain a portion of ReLUs** to preserve accuracy. AutoFHE uses EvoReLUs to replace **all ReLUs** while preserving accuracy. In principle, AutoFHE can *implicitly* remove any unnecessary ReLUs. For instance, AutoFHE may approximate some ReLUs with a linear function as part of the optimization, which is equivalent to removing the corresponding ReLU. *Explicitly* combining these two research directions is an interesting direction for future work to improve the efficiency of secure inference in both FHE and secure MPC. But this is beyond the scope of the current paper.
>
> > **Q3:** If EvoReLU does not help in reducing latency, then is it replaced (from AppReLU) to improve accuracy? Or there is something I am missing? Does EvoReLU introduces polynomial error based on layer sensitivity? Please clarify, as the authors had mentioned Approx ReLU uses similar approx error for all ReLU layers.
>
> **A3:** There seems to be a misunderstanding. We **DID NOT** use AppReLU in our method. Both the proposed EvoReLU and AppReLU use the framework of composite polynomials for approximating ReLUs; the optimization of the polynomial degrees and coefficients is entirely different. The FHE-MP-CNN baseline uses the same AppReLU for all layers, specifically the same polynomial function and the same coefficients. AppReLU's coefficients are obtained via function (sign function in this case) approximation given a multiplicative depth constraint. The proposed AutoFHE jointly optimizes the polynomial degrees and their coefficients of *all* EvoReLUs in the network to directly maximize the accuracy as well as minimize the number of bootstrapping operations. Due to the varying sensitivity of different layers, AutoFHE can assign *different EvoReLUs to each layer* while preserving accuracy.  In Figure 12 and Figure 13 (see the updated version), we plot EvoReLUs layer by layer for ResNet-56. We can observe that some layers prefer high-precision EvoReLUs, and others can use low-degree smooth functions.
>
> > **Q4:** If you replace the EvoReLU with ReLU during backward pass, wont it introduce significant error? How that error is compensated in the framework? Why EvoReLU introduce exploding gradient issue, and why it can't be solved by traditional mean of solving exploding gradients? STE was introduced to primarily deal with non differentiable functions, not to handle exploding gradients. So, the citation of STE in this context is confusing!
>
> **A4:** Because we use polynomials to replace every ReLU. The neural network can be regarded as a very high-degree polynomial function. So, the problem of gradient explosion is more severe than in ReLU networks. Previous literature, like [1], [2], also observed this problem, although they use low-degree polynomials. The proposed polynomial-aware training (PAT) will not lead to large errors because EvoReLU can approximate the ReLU with relatively high precision. STE and QAT use **different** functions for forward and backward propagation, so we cited them due to the similarity of our PAT. We respectfully disagree with the reviewer’s assertion that techniques motivated in one context cannot be helpful in other contexts. We believe the concept of STE can be useful beyond non-differential functions to functions that may also suffer from gradient explosion.
>
> ### Reference:
>
> [1] Delphi. USENIX Security Symposium, 2020.
>
> [2] Safenet. ICLR 2020.

---

> ### Author Response · Authors · 2022-11-16
> **Response to Reviewer b7YQ (Part II)**
>
> >**Q5:** MoCoEv and Sec. 3.3 are written in a hurried way without properly mentioning the inspiration of the proposed solution. I find no discussion on possible alternatives and why we should buy this "R-CCDE" as the best potential solution.
>
> **A5:** We added a case study in section 1 to help readers better understand our motivation. There are two possible alternative solutions for optimizing composite polynomials. One is the Minimax polynomials used by our baseline FHE-MP-CNN. Another alternative is to use a gradient-based optimizer, but it is not stable due to exploding gradients. This motivated us to seek a gradient-free optimizer. DE only uses the difference between solutions to optimize them. We found composite polynomials, i.e., $f(x)=f_3(f_2(f_1(x)))$, are well-suited for co-evolution because there is a natural grouping of the variables. We explained all details in Appendix C and D (see updated version). Lastly, we do not claim that *R-CCDE* is the *best possible solution*. We demonstrate that R-CCDE is very effective. We remind the reviewer that our broader contribution is a framework for adapting CNNs for evaluation over FHE. R-CCDE is one part of the solution, albeit an important one.
>
> > **Q6:** There is no pareto frontier of the search space, either for bootstrap or the EvoReLU polynomial choices. There is no latency vs accuracy plot either.
>
> **A6:** Perhaps the reviewer missed them, but the requested plots are already present in the paper. We plot the Pareto fronts of accuracy and bootstrapping, accuracy, and inference time (see Figure 3, Figure 4, Figure 6, and Figure 7 in the original submission). You can also find them in Figure 1, Figure 5, Figure 6, and Table 2 in our updated version. Maybe the reviewer meant Pareto front of the **search space**? The search space is extremely large ($10^{79}-10^{230}$), and we do not have the *ground truth* solution. So, we demonstrate the efficacy of AutoFHE by comparing it with the state-of-the-art solution.
>
> > **Q7:** Details of latency improvement computation is missing (the one mentioned as $17\%$ and $22\%$).
>
> **A7:** This is relatively straightforward, $100 \times \frac{T_{old} - T_{new}}{T_{old}}$.
>
> > **Q8:** Additional results on higher resolution datasets are missing. Both CIFAR-10 and 100 are of size 32x32x3.
>
> **A8:** We followed existing work under FHE, like LoLa [3], HEMET [4], and FHE-MP-CNN to use CIFAR to benchmark performance on "encrypted" data, CIFAR is a relatively high-resolution dataset that can be practically evaluated now. The resolution of CIFAR images is $3\times32\times32$. We set the CKKS polynomial degree to $N=2^{16}$ to have enough slots for all items of tensors. The memory complexity is $\mathcal{O}(N)$ and the computational complexity of multiplication is $\mathcal{O}(N\log N)$. When we set $N=2^{16}$, its practical memory footprint is about 500 GB. If we use ImageNet with a resolution of $3\times 224 \times 224$, we may need $49\times$ larger memory, and the inference time will be much slower. To realize the goal of practically processing high-resolution images in the encrypted domains, we need advancements in RNS-CKKS primitives, custom hardware for FHE, more efficient packing algorithms, etc. This is the reason why all existing works use CIFAR to benchmark performance. Although we cannot use CNNs to classify high-resolution images due to the constraints mentioned above of FHE, we provided an experimental result of AutoFHE on plaintext ImageNet in Appendix G (see the updated version). The experiment shows the proposed AutoFHE can trade off the accuracy and the depth consumption. This experiment demonstrates that AutoFHE can design a network that can be evaluated in “ciphertext” once the above practical limitations can be resolved.
>
> > **Q9:** The authors should also compare with more advanced search methods, instead of only comparing with NSGA-II, which is not necessarily the state-of-the-art.
>
> **A9:** NSGA-II is widely and very effectively used in the neural architecture search (NAS) community as a multi-objective search algorithm, e.g., [5], [6]. When we compare our method with NSGA-II in an ablation study, we demonstrate the effectiveness of **co-evolution**. If you have concrete suggestions for *more advanced search algorithms*, we will be happy to consider them.
>
> > **Q10:** Please clarify (in the preliminaries) why bootstrapping operation is costly in a bit more details.
>
> **A10:** Practical implementations of bootstrapping involve five stages, including ModRaise, CoeffToSlot, EvalMod (approximate evaluation of $mod$ function), and SlotToCoeff. These operations involve a lot of homomorphic multiplications and rotations, both of which are costly operations, especially the latter.
>
> ### Reference:
>
> [3] Low latency privacy preserving inference. ICML 2019.
>
> [4] Hemet. ICML 2021.
>
> [5] Multi-objective evolution for Generalizable Policy Gradient Algorithms. arXiv 2022.
>
> [6] NSGANetV2, ECCV 2020

---

> ### Comment · Reviewer_b7YQ · 2022-11-22
> **Thanks and further comments**
>
> Dear Authors,
>
> Apologies for delayed response! I have read the rebuttal response. I have the following questions on your response:
>
> 1. As NSGA-III and U-NSGA-III [3] are also multi and many objective co-evaluation methods, it would good to compare with them with that of MoCoEv.
>
> 2. There is a domain called pruning, that tries to minimize the number of weights given a pre-defined constrained. It uses various non-convex optimization example ADMM, I wonder why such methods wont work for this problem statement.
>
> 3. You should also be aware of various Neural Architecture Search (NAS) methods that often require multi- many objective optimizations, that could have been used to solve such MoCoEv problems.
>
> 4. The incorporation of [1][2] should have made the problem easier, as one of the motivation was to figure out the layers, requiring more or less polynomial approximation of ReLUs, that these methods can do.
>
> 5. Fig. 1 has four circles, but in the legend only three are there.
>
> Let me be very clear: **to my understanding this paper proposes an efficient optimization to yield latency efficient FHE models, thus I want to understand and evaluate this manuscript's contributions with respect to the existing or potential alternatives ML community already has**.
>
>
> [1] selective network linearization, ICML 2022.
>
> [2] DeepReDuce, ICML 2021.
>
> [3] U-NSGA-III: A Unified Evolutionary Algorithm for Single, Multiple, and Many-Objective Optimization (https://www.egr.msu.edu/~kdeb/papers/c2014022.pdf).

---

> > ### Author Response · Authors · 2022-11-22
> > **A Response to Follow-Up Questions of Reviewer b7YQ (4/4)**
> >
> > ### Reference:
> >
> > [1] selective network linearization, ICML 2022.
> >
> > [2] DeepReDuce, ICML 2021.
> >
> > [3] U-NSGA-III: A Unified Evolutionary Algorithm for Single, Multiple, and Many-Objective Optimization
> >
> > [4] Multi-objective evolution for Generalizable Policy Gradient Algorithms. arXiv preprint arXiv:2204.04292, 2022.
> >
> > [5] Neural Architecture Transfer, IEEE Transaction on Pattern Analysis and Machine Intelligence (PAMI), 2021.
> >
> > [6] NSGANetV2:Evolutionary Multi-Objective Surrogate-Assisted Neural Architecture Search, ECCV 2020
> >
> > [7] SM-NAS: Structural-to-Modular Neural Architecture Search for Object Detection, AAAI 2020
> >
> > [8] NSGA-NET: Neural Architecture Search using Multi-Objective Genetic Algorithm, GECCO 2019
> >
> > [9] Efficient Multi-Objective Neural Architecture Search via Lamarckian Evolution, ICLR 2019
> >
> > [10] Hemet: A homomorphic-encryption-friendly privacy-preserving mobile neural network architecture. ICML 2021.
> >
> > [11] Low-complexity deep convolutional neural networks on fully homomorphic encryption using multiplexed parallel convolutions. ICML 2022.

---

> > ### Author Response · Authors · 2022-11-22
> > **A Response to Follow-Up Questions of Reviewer b7YQ (3/4)**
> >
> > > **Q4.** The incorporation of [1][2] should have made the problem easier, as one of the motivation was to figure out the layers, requiring more or less polynomial approximation of ReLUs, that these methods can do. Let me be very clear: to my understanding this paper proposes an efficient optimization to yield latency efficient FHE models, thus I want to understand and evaluate this manuscript's contributions with respect to the existing or potential alternatives ML community already has.
> >
> > **A4.** Thanks for sharing those references. The cryptographic scheme in the suggested paper is secure MPC. The cryptographic scheme in our paper is FHE (RNS-CKKS). These are **fundamentally different schemes** and have completely different constraints. **Solutions developed for secure MPC cannot be directly employed for RNS-CKKS without significant modifications.** Below we describe why using the suggested papers as examples.
> >
> > Secure MPC can evaluate ReLU exactly but is a costly operation. **So the references [1] and [2] suggested here seek to reduce the number of ReLUs, but not entirely replace them with polynomials as in our paper.** For secure inference using secure MPC one needs to **retain some ReLUs** to preserve accuracy. If most of ReLUs are removed, a dramatic drop in accuracy is observed. Furthermore, since secure MPC can operate on each individual element of a vector at a time, [1] seeks to eliminate ReLU for some elements of the pre-activation tensor. So both [1] and [2] retain some ReLU operations, which will still need to be approximated with high-precision for RNS-CKKS.
> >
> > Both the aforementioned properties are not sufficient for FHE since RNS-CKKS exerts additional constraints.
> >
> > 1. RNS-CKKS is based on SIMD encoding of the data, i.e., data is encoded into a vector and arithmetic operations act on **all** elements of a vector at once. So one cannot manipulate an individual element of such a vector without additional homomorphic multiplications and rotations which are very costly. So the solution in [1] where ReLU is selectively applied only to some elements of the vector is not feasible/efficient in RNS-CKKS.
> > 2. The need for bootstrapping in RNS-CKKS places additional constraint since one has to be careful to ensure that pair of ciphertexts on which arithmetic operations (additions or multiplications) are being performed should ideally be at the same *multiplicative level*, else *levels* will be wasted and there will not be any latency gains. So, without co-designing the ReLU approximation and the homomorphic evaluation circuit, any proposed solution is not likely to yield real latency gains without significant performance loss.
> >
> > These constraints render the direct application of the solutions in [1] and [2] completely ineffective for RNS-CKKS. For example, Eq. 3 in [1] does not account for approximation of ReLUs that are retained, the optimal placement of bootstrapping operations, nor the multi-objective nature of our formulation. Therefore, the solutions in [1] and [2] have to be redesigned from the ground up for application to RNS-CKKS. All existing solutions including [10],[11], for FHE instead approximate **all** ReLUs in the network with polynomials.
> >
> > We believe that the concept of ReLU reduction and polynomial approximation of ReLUs are complementary and both can be employed for secure inference using RNS-CKKS. This is an exciting line of research. However, development and validation of the utility of such a solution is a research project by itself and is beyond the scope of this paper.
> >
> > In principle, AutoFHE can also remove ReLU since degree=1, which is equivalent to removing ReLU. Empirically we observe that the solutions found by AutoFHE can go degree=2.
> >
> > > **Q5.** Fig. 1 has four circles, but in the legend only three are there.
> >
> > **A5.** The plot has 8 circles, four for AutoFHE solutions and four for FHE-MP-CNN solutions. Please note that the circles are of different size, and size of the circles corresponds to the number of bootstrapping operations. The legend shows three examples of illustrate the relationship between the size of circles and the number of bootstrapping operations. The exact number of bootstrapping operations for each of the eight solutions can be found in Table 2.

---

> > ### Author Response · Authors · 2022-11-22
> > **A Response to Follow-Up Questions of Reviewer b7YQ (2/4)**
> >
> > > **Q2.** There is a domain called pruning, that tries to minimize the number of weights given a pre-defined constrained. It uses various non-convex optimization example ADMM, I wonder why such methods wont work for this problem statement.
> >
> > **A2.** We are little confused by this comment. Pruning is typically employed for minimizing the weights of convolution/linear layers. That goal is orthogonal/complementary to the goals of this paper, which is to find FHE friendly polynomial approximations of activation functions (ReLU). In this case, we respectfully disagree that weight pruning methods are relevant to the goals of our paper.
> >
> > If the reviewer meant applying the pruning idea to polynomial approximations of ReLU, then we are afraid this is not straightforward and needs significant new methods to solve our problem. Firstly, we have two objectives to optimize, where the second objective, number of bootstrapping ops, is discrete and not differentiable. Any ADMM-based solution needs to optimize both objectives which is not straightforward without gradients. Also ADMM needs to be executed multiple times, once for each combination of the two objectives into a single objective in order to obtain a Pareto-front of solutions. Secondly, as we show in Figure 2 of the updated paper, simply reducing the degree of the polynomials may not provide any latency benefits since it may not translate to fewer bootstrapping operations, and thus no latency improvements. So one cannot ignore our second objective when using ADMM. Thirdly, ADMM type algorithms for pruning, need gradients, which as the research community observed in multiple papers, polynomial ReLU approximations lead to exploding gradients. So the ADMM suggestion needs significant adaptations to work for our problem, and there is still no guarantee that such a hypothetical solution can outperform our proposed solution. We believe this is an interesting line of research, but is whole research project in itself.
> >
> >
> > > **Q3.** You should also be aware of various Neural Architecture Search (NAS) methods that often require multi- many objective optimizations, that could have been used to solve such MoCoEv problems.
> >
> > **A3.** We are certainly aware of the many papers that employ multi-objective optimization for NAS. For example, [4-9] below and possibly more. Most of them either employ NSGA-II, Lamarckian Evolution or MOEA/D. However, in *all* of these cases, the number of optimization variables is of the order of 20-40 variables. So there is no need to employ co-evolution (please refer to A1 for why this is the case). Multi-Objective NAS typically defines a search space and employs multi-objective optimization. In that spirit, AutoFHE also can be considered multi-objective NAS. We define a search space and employ multi-objective optimization. The difference being that our search space is completely different, our objectives are completely different and our solution is based on co-evolution due to large number of variables. We did not refer to our approach as NAS, since our goal is not to searching for architectures, but are rather searching for polynomial approximations of a specific type of layer (ReLU in this case).

---

> > ### Author Response · Authors · 2022-11-22
> > **A Response to Follow-Up Questions of Reviewer b7YQ (1/4)**
> >
> > We really appreciate your time and efforts to suggest alternative solutions to improve and compare with our approach. We would like to reiterate that alternative, and possibly better, approaches may exist to solves our proposed optimization problem (Eq. 3 in paper). However, the existence of such methods does not diminish the effectiveness of the proposed approach. We believe that our approach already provides **significant** improvements for the problem we are addressing (see Fig. 1), and other alternative approaches may only serve to improve the results.
> >
> > Below we provide our concrete comments on your suggestions. Unfortunately, none of the suggestions can be directly employed for our problem, and adapting them for our problems needs significant efforts are research projects in their own right and certainly opens exciting new solution paths. We will be happy to explore them in future work, perhaps even in collaboration with the reviewer if he/she wishes.
> >
> > > **Q1.** As NSGA-III and U-NSGA-III [3] are also multi and many objective co-evaluation methods, it would good to compare with them with that of MoCoEv.
> >
> > **A1.** Thanks for suggesting NSGA-III and its generalized version U-NSGA-III. We would like to first clarify the exact difference between MOCoEv and the NSGA-II baseline in our paper. Then we describe a couple of reasons why both the suggestions provided by the reviewer are not directly suitable for our problem.
> >
> > **Difference Between MOCoEv and NSGA-II Baseline:** As a reminder, the dimensionality of our search space is high (114 ~ 330) and our search space is very large ($10^{79}\sim10^{230}$). Empirically it has been observed that vanilla NSGA-II is effective up to ~30 variables. Co-evolution is typically employed for dealing with large number of variables by partitioning the variables into groups of smaller number of variables. For example, 300 variables are grouped into 10 groups of 30 variables each. Co-evolution evolves **one group** at a time as opposed to evolving **all** variables at the same time. **MOCoEv groups the variables and evolves one group at a time.** But MOCoEv still uses the concept of non-domination and crowding distance to select *diverse* individuals to retain during search, which are the core concepts of NSGA-II. The baseline NSGA-II we consider operates directly on **all** the variables without grouping, and still employs non-domination and crowding distance. In summary MOCoEv can be considered as applying NSGA-II to groups of variables and iterating through the groups as opposed to all variables.
> >
> > **Why NSGA-III and U-NSGA-III are not suitable:**
> >
> > 1. NSGA-III is specifically designed for **three or more** objectives (in practice it can work well up to 15 objectives). Our problem, on the other hand has **two objectives**. NSGA-II is specifically designed for **two objectives** and is better suited in our context. NSGA-III and U-NSGA-III *can be* adapted to solve two objective problems, but it needs custom adaptation and there is no obvious way to do it. Furthermore, as Table 3 in [3] (U-NSGA-III) shows, when comparing NSGA-II, NSGA-III, and U-NSGA-III, there is relatively minor difference  between NSGA-II and adapted NSGA-III and U-NSGA-III. For ZDT3 and ZDT6 problems, **NSGA-II outperforms NSGA-III and U-NSGA-III**. Most of the experimental results show that **their hypervolume values are very close**. However, using co-evolution MOCoEV improves the hypervolume by x1.32 and x1.24 over ResNet-20 and ResNet-32 compared with NSGA-II (see Table 4 in our paper).
> > 2. NSGA-III and U-NSGA-III are both reference based search methods. MOEA/D is another such reference based method.  Instead of using crowding distance, as is done in NSGA-II, to select diverse individuals, reference based methods use *user-specified* reference lines. This is because the concept of crowding distance is not well-defined for three or more objectives. So, to use NSGA-III we need to specify reference lines which is not straightforward, as one of our objectives is discrete (number of bootstrapping ops) while the other is continuous (accuracy).
> >
> > Due to the above reasons we do not believe NSGA-III or U-NSGA-III is better suited than NSGA-II for our problem. Having said that, there are no barriers to adopting NSGA-III or U-NSGA-III within MOCoEv since the main idea of MOCoEv is to combine co-evolution (grouping) with multi-objective evolutionary optimization, and the multi-objective part can be replaced by NSGA-III or U-NSGA-III if one desires. But we do not believe that there will be a significant difference in the final conclusions of our paper.

---

> > ### Author Response · Authors · 2022-12-06
> > **A Gentle Reminder to Reviewer b7YQ**
> >
> > We greatly appreciate your time and efforts. We thank you for taking the time to read our original response and the updated version of our paper. We replied to your follow-up questions. Please see ["A Response to Follow-Up Questions of Reviewer b7YQ(1/4)-(4/4)"](https://openreview.net/forum?id=Hq16Jk2bVlp&noteId=gm04q-j8SpC). As the discussion deadline is approaching, we are wondering if you had a chance to read our response to your follow-up questions. Did our response address your concerns? We look forward to your reply. Thanks a lot.

---

> > ### Author Response · Authors · 2022-12-08
> > **Look Forward to Replying from Reviewer b7YQ**
> >
> > This is a gentle reminder because the discussion window is being closed in the coming days. We are wondering if you had a chance to read our response to your follow-up questions. Did our response address your concerns?
> >
> > **We greatly appreciate your time, efforts and comments. We look forward to your reply. Thanks a lot.**

---

> ### Author Response · Authors · 2022-12-10
> **Look Forward to Your Reply**
>
> Dear Reviewer b7YQ,
>
> As the discussion is ending soon, we are writing this gentle reminder to ask if you had a chance to read our response to your follow-up questions, say "A Response to Follow-Up Questions of Reviewer b7YQ"  [(1/4)](https://openreview.net/forum?id=Hq16Jk2bVlp&noteId=gm04q-j8SpC), [(2/4)](https://openreview.net/forum?id=Hq16Jk2bVlp&noteId=TSMJiKYJMK), [(3/4)](https://openreview.net/forum?id=Hq16Jk2bVlp&noteId=E4-kovTKg0) and [(4/4)](https://openreview.net/forum?id=Hq16Jk2bVlp&noteId=tOXhBI0h8b). Did our response address your concerns? We are happy to answer your other questions.
>
> **We greatly appreciate your time, efforts, and comments. We look forward to your reply. Thanks very much.**
>
> Author

---

### Official Review · Reviewer_Lx4T · 2022-10-24

**Confidence:** 4
**Correctness:** 4
**Technical Novelty And Significance:** 4
**Empirical Novelty And Significance:** 4
**Recommendation:** 6

**Clarity, Quality, Novelty And Reproducibility:**

Great quality, very clear, self-contained, and original work with novel techniques.

**Strength And Weaknesses:**

Strength: this paper provides a framework for secure CNN inference that you could solely rely on it. The proposed techniques are novel and sound. And the experiments and evaluations are complete. This paper is also clearly written and provides all the necessary details.

My one question is about the choice of approximation functions. AFAIK Chebyshev approximations are quite generic. Is there any room for improvement given that we are particularly interested in the sign function? (Similar problem is considered here https://ia.cr/2021/572 but for sine function). Or maybe it will bring more complexity in designing the later search algorithm.

**Summary Of The Paper:**

This paper provides a novel and comprehensive framework for convolutional neural network inference over a fully homomorphic encryption environment. Based on previous work FHE-MP-CNN, they made significant improvements by
1) using a new approximation for the non-linear activation function ReLU
2) an automatic procedure that helps choose your approximation function which simultaneously maximizes the accuracy and minimizes the computation cost.
3) a new training procedure that better suits the proposed inference framework.

**Summary Of The Review:**

I think it's a great paper that matches this conference. Shall definitely be accepted.

---

> ### Author Response · Authors · 2022-11-16
> **Response to Reviewer Lx4T**
>
> ### We thank you for your positive and valuable comments. To our knowledge, this is the first work to automatically reduce the time-consuming bootstrapping operations and maximize accuracy.
>
> > **Q:** My one question is about the choice of approximation functions. AFAIK Chebyshev approximations are quite generic. Is there any room for improvement given that we are particularly interested in the sign function? (Similar problem is considered here https://ia.cr/2021/572 but for sine function). Or maybe it will bring more complexity in designing the later search algorithm.
>
> **A:** This is a great suggestion, and we thank you for sharing that paper. There are many possible bases for function approximation, and we mentioned a few similar articles when we introduced approximation of the modular reduction function of bootstrapping in preliminaries. We believe the choice of polynomial bases for approximation should be guided by two considerations, (i) quality of approximation (low error) and (ii) ease of training (vanishing or exploding gradients). We chose Chebyshev bases for two reasons, (i) because it is bounded within [-1, 1] and can alleviate the gradient exploding problem, which is necessary for our polynomial aware training (PAT) to be effective, and (ii) using generic bases allows us to be agnostic to the particular activation function used in the network. We agree that some bases may be better suited for some activation functions, and further gains can be obtained. Note that our proposed approach is itself agnostic to which polynomial bases are utilized, i.e., we can replace Chebyshev with other bases functions without adding extra complexity to our search algorithm. Exploring the best choice in terms of precision of approximation, ease of training, and depth consumption is an interesting topic for future exploration.

---

> ### Author Response · Authors · 2022-12-08
> **A Gentle Reminder to Reviewer Lx4T**
>
> This is a gentle reminder to ask if you have a chance to read our rebuttal, including [Response to Reviewer Lx4T](https://openreview.net/forum?id=Hq16Jk2bVlp&noteId=mifjJDPMiL) and [An Updated Version of AutoFHE Paper](https://openreview.net/forum?id=Hq16Jk2bVlp&noteId=3GUOWFJTk4). If you want us to clarify any other points, please let us know.
>
> **We greatly appreciate your time, efforts and comments. We look forward to your reply. Thanks a lot.**

---

> ### Author Response · Authors · 2022-12-10
> **Look Forward to Your Reply**
>
> Dear Reviewer Lx4T,
>
> As the discussion is ending soon, we are writing this gentle reminder to ask if you had a chance to read our response, including [Response to Reviewer Lx4T](https://openreview.net/forum?id=Hq16Jk2bVlp&noteId=mifjJDPMiL) and [An Updated Version of AutoFHE Paper ](https://openreview.net/forum?id=Hq16Jk2bVlp&noteId=3GUOWFJTk4). Do you need more discussion?
>
> **We greatly appreciate your time, efforts, and comments. We look forward to your reply. Thanks very much.**
>
> Author

---

### Official Review · Reviewer_Btk1 · 2022-10-25

**Confidence:** 4
**Correctness:** 2
**Technical Novelty And Significance:** 2
**Empirical Novelty And Significance:** 2
**Recommendation:** 5

**Clarity, Quality, Novelty And Reproducibility:**

Clarity of the paper is lacking is many cases. Please note the weaknesses section in the review. Though an interesting solution is proposed efficacy of the novelty could not be evaluated from the provided experimental evaluation. Reproducibility could be improved.

**Details Of Ethics Concerns:**

No ethical concerns as of yet.

**Strength And Weaknesses:**

Strengths::
The paper proposes an interesting concept.

Weaknesses::
The paper can be improved based on the following points.
1) It might be better to provide a motivation case study to show the shortcomings of the design choices of the state-of-the-art approach FHE-MP-CNN in section 1. This would motivate the readers to understand the importance of the proposed mechanism.
2) In section 3, it is mentioned, "We directly optimize the end-to-end objective to facilitate finding the optimal combination of layerwise polynomials." However, from the text it is not evident how the design choices were made nor how layer wise polynomials were optimised. Please specify.
3) In section 3, please specify what each variables such as x, p, d, y means such that readers are able to understand the concept without domain expertise on the particular topic. This is to improve readability.
4) Please briefly specify why CIFAR datasets were only utilised for evaluation in section 4. Why not other datasets? How will the proposed mechanism perform with other datasets (if such an evaluation is done or at least discuss the limitations of the proposed).
5) Experimental section (section 4) lacks comparative study with state-of-the-art methods to show the efficacy of the proposed. Please provide such comparative studies or discuss the evaluations.

**Summary Of The Paper:**

In this paper, the authors propose Automated adaption of CNNs for evaluation over fully homomorphic encryption (FHE) called AutoFHE to accelerate CNNs on FHE and automatically design a homomorphic evaluation architecture. AutoFHE claims to exploit the varying sensitivity of approximate activations across different layers in a network and jointly evolves polynomial activations (EvoReLUs) and searches for placement of bootstrapping operations for evaluation under RNS-CKKS.

**Summary Of The Review:**

1) It might be better to provide a motivation case study to show the shortcomings of the design choices of the state-of-the-art approach FHE-MP-CNN in section 1. This would motivate the readers to understand the importance of the proposed mechanism.
2) In section 3, it is mentioned, "We directly optimize the end-to-end objective to facilitate finding the optimal combination of layerwise polynomials." However, from the text it is not evident how the design choices were made nor how layer wise polynomials were optimised. Please specify.
3) In section 3, please specify what each variables such as x, p, d, y means such that readers are able to understand the concept without domain expertise on the particular topic. This is to improve readability.
4) Please briefly specify why CIFAR datasets were only utilised for evaluation in section 4. Why not other datasets? How will the proposed mechanism perform with other datasets (if such an evaluation is done or at least discuss the limitations of the proposed).
5) Experimental section (section 4) lacks comparative study with state-of-the-art methods to show the efficacy of the proposed. Please provide such comparative studies or discuss the evaluations.

---

> ### Author Response · Authors · 2022-11-16
> **Response to Reviewer Btk1**
>
> ### We thank your efforts and constructive suggestion. We followed your advice to improve our paper (see the updated version).
>
> > **Q1:** It might be better to provide a motivation case study to show the shortcomings of the design choices of the state-of-the-art approach FHE-MP-CNN in section 1. This would motivate the readers to understand the importance of the proposed mechanism.
>
> **A1:** We added a case study in Section 1 to help readers to understand our motivation better (see the updated version).
>
> > **Q2:** In section 3, it is mentioned, "We directly optimize the end-to-end objective to facilitate finding the optimal combination of layerwise polynomials." However, from the text, it is not evident how the design choices were made nor how layer-wise polynomials were optimised. Please specify.
>
> **A2:** The motivation, "direct optimization of end-to-end objectives", is addressed from two aspects: 1) We design the search objective (see Eq. 3) to directly maximize the classification accuracy and minimize the number of bootstrapping operations. 2) To optimize the search objective (i.e., Eq. 3), we propose MOCoEv, a multi-objective search algorithm to explore the Pareto front by optimizing the end-to-end objectives, i.e., classification accuracy and the number of bootstrapping operations. To this end, MOCoEv searches for layerwise degrees of all EvoReLU polynomials. Coefficients of layerwise EvoReLUs are obtained by maximizing the classification accuracy (see Eq. 3). Finally, we use polynomial-aware training (PAT) to fine-tune learnable weights of networks to adapt to layerwise EvoReLUs. PAT also minimizes the end-to-end objective, which is cross-entropy loss.
>
> > **Q3:** In section 3, please specify what each variable such as x, p, d, y means such that readers are able to understand the concept without domain expertise on the particular topic. This is to improve readability.
>
> **A3:** We explained these variables in Section 3.1. $x\in[-1,1]$ and $y\in[0,1]$ are input and output variables of EvoReLU. $p^d(x)$ is a composite polynomial function with degree $d$. In Appendix A (see the updated version), we explained variable notations to clarify it.
>
> > **Q4:** Please briefly specify why CIFAR datasets were only utilised for evaluation in section 4. Why not other datasets? How will the proposed mechanism perform with other datasets (if such an evaluation is done or at least discuss the limitations of the proposed).
>
> **A4:** To benchmark performance on "encrypted" data, CIFAR is a relatively high-resolution dataset that can be practically evaluated now (see LoLa [1], HEMET [2], and FHE-MP-CNN [3]). The resolution of CIFAR images is $3\times32\times32$. We set the CKKS polynomial degree to $N=2^{16}$ to have enough slots for all items of tensors. The memory complexity is $\mathcal{O}(N)$, and the computational complexity of multiplication is $\mathcal{O}(N\log N)$. When we set $N=2^{16}$, its practical memory footprint is about 500 GB. If we use ImageNet with a resolution of $3\times 224 \times 224$, we may need $49\times$ larger memory, and the inference time will also be much slower. To realize the goal of practically processing high-resolution images in the encrypted domains, we need advancements in RNS-CKKS primitives, custom hardware for FHE (FYI, this is already being developed by many companies, including Intel), more efficient packing algorithms, etc. This is the reason why all existing works use CIFAR to benchmark performance. Although we cannot use CNNs to classify high-resolution images due to the constraints mentioned above, we provided experimental results of AutoFHE on plaintext ImageNet in Appendix G (see the updated version). We estimate the accuracy of plaintext ImageNet images since FHE encryption and polynomial approximation errors are orthogonal, and count depth consumption as the inference cost under RNS-CKKS. The experiment shows that AutoFHE can trade off the accuracy and the depth consumption. This experiment demonstrates that AutoFHE can design a network that can be evaluated in “ciphertext” once the above practical limitations can be resolved.
>
> > **Q5:** Experimental section (section 4) lacks comparative study with state-of-the-art methods to show the efficacy of the proposed. Please provide such comparative studies or discuss the evaluations.
>
> **A5:** FHE-MP-CNN[3] is the most recent and state-of-the-art baseline for **deep** CNNs using **bootstrapping** with **high accuracy** (>90% on CIFAR-10). And we did compare our method with FHE-MP-CNN in the paper.
>
> > **Q6:**  Reproducibility could be improved.
>
> **A6:** Check the anonymous repository of our code.
>
> ### Reference:
> [1] Low latency privacy preserving inference. ICML 2019.
>
> [2] Hemet: A homomorphic-encryption-friendly privacy-preserving mobile neural network architecture. ICML 2021.
>
> [3] Low-complexity deep convolutional neural networks on fully homomorphic encryption using multiplexed parallel convolutions. ICML 2022.

---

> ### Author Response · Authors · 2022-12-06
> **A Gentle Reminder to Reviewer Btk1**
>
> We greatly appreciate your time and efforts. We replied to your questions and followed your suggestions to improve our paper. As the discussion deadline is approaching, we are wondering if you had a chance to read our response and the updated version of the paper. Did our response address your concerns? We look forward to your reply and will answer your follow-up questions if you have any. Thanks a lot.

---

> ### Author Response · Authors · 2022-12-08
> **Look Forward to Replying from Reviewer Btk1**
>
> As the discussion window is being closed in a few days, we are writing this gentle reminder. We hope you can have a chance to read our [Response to Reviewer Btk1](https://openreview.net/forum?id=Hq16Jk2bVlp&noteId=h8HDgm6scar). We followed your advice to include a motivation case study in our updated version of the paper. We also discussed the dataset problem and added a search experimental result on ImageNet (Please see [An Updated Version of AutoFHE Paper ](https://openreview.net/forum?id=Hq16Jk2bVlp&noteId=3GUOWFJTk4)). We also showed better performance by simply changing the learning rate and the number of fine-tuning epochs.
>
> **We greatly appreciate your time, efforts and comments. We look forward to your reply. Thanks a lot.**

---

> ### Author Response · Authors · 2022-12-10
> **Look Forward to Your Reply**
>
> Dear Reviewer Btk1,
>
> As the discussion is ending soon, we are writing this gentle reminder to ask if you had a chance to read our response, including [Response to Reviewer Btk1](https://openreview.net/forum?id=Hq16Jk2bVlp&noteId=h8HDgm6scar) and [An Updated Version of AutoFHE Paper ](https://openreview.net/forum?id=Hq16Jk2bVlp&noteId=3GUOWFJTk4). We answered your questions and included your suggestions in our updated paper. Did our response address your concerns? We are happy to answer your follow-up questions.
>
> **We greatly appreciate your time, efforts, and comments. We look forward to your reply. Thanks very much.**
>
> Author

---

### Author Response · Authors · 2022-11-16
**An Updated Version of AutoFHE Paper**

### We greatly appreciate all reviewers' efforts and constructive comments. We submitted an updated version of our paper.

**1.** We updated the results obtained by AutoFHE by changing two hyperparameters. We changed two hyperparameters (learning rate and epochs) in the polynomial-aware training (PAT). We changed the learning rate from $8\times10^{-5}$ and $5\times10^{-5}$ to $5\times10^{-4}$ and $2\times10^{-4}$ for CIFAR-10 and CIFAR-100, respectively. We set finetuning epoch(s) to **one** during the search, and **ten** after the search. These simple changes lead to significant performance gains. We updated the paper with the new experimental results; please see **Figure 1**, **Figure 5**, **Figure 6**, and **Table 2**. Experimental results on CIFAR-10 indicate that in comparison to the state-of-the-art solution, AutoFHE can reduce inference time (50 images on 50 CPU threads) by up to **3,297 seconds** (**43%**) while preserving the accuracy (92.68%). AutoFHE also improves the accuracy of ResNet-32 on CIFAR-10 by up to **0.48%** while accelerating inference by 382 seconds (7%).

**2.** Reviewer Btk1 and b7YQ mentioned additional datasets. We followed the most recent existing work on FHE to use CIFAR-10/-100 to benchmark AutoFHE. Currently, *ciphertext* high-resolution images cannot be practically processed under RNS-CKKS due to the constraints of RNS-CKKS primitives, hardware limitations, implementation challenges, and packing algorithms. For example, multiple terra bytes of memory is necessary when image resolution increases. These problems are orthogonal to the goals of this paper. However, we added experimental results on *plaintext* ImageNet in Appendix G (see the updated version). We estimate accuracy on plaintext data and use depth consumption as the inference cost on FHE. This preliminary experiment shows AutoFHE can effectively trade off accuracy and depth consumption on large-scale high-resolution image datasets.

---

### Author Response · Authors · 2022-11-18
**A Gentle Reminder to all Reviewers**

We appreciate all reviewers' time, efforts, and constructive comments. This is a gentle reminder to ask if you have a chance to read our rebuttal. Do our responses address your concerns? As the discussion deadline is approaching, we look forward to your reply and are happy to answer your follow-up questions. Thanks a lot.

---

### Decision · Program_Chairs · 2023-01-20

**Decision:**

Reject

**Justification For Why Not Higher Score:**

The case for this paper is very borderline as discussed in the AC-Reviewer meeting note. For the security community there are some interesting solid empirical findings that would encourage discussion among researchers. From ML perspective, the question on whether the search / optimization method utilized by AutoFHE is novel and different from the already existing methodology still remains.

Given that no reviewer showed strong support for acceptance (supporting reviewer decreased score after AC-Reviewer discussion),  AC encourages the authors to revise the paper with evaluation beyond CIFAR and also improvement on clarity.


**Justification For Why Not Lower Score:**

N/A

**Metareview: Summary, Strengths And Weaknesses:**


The paper proposes automated adaption of convolutional neural networks for evaluation over fully homomorphic encryption (AutoFHE). The main problem the authors tackle is secure inference of deep neural networks which was recently demonstrated under RNS-CKKS fully homomorphic encryption (FHE) scheme.  AutoFHE utilizes the varying sensitivity of approximate activations across different layers in a network and jointly evolves polynomial activations (EvoReLUs) and searches for placement of bootstrapping operations for evaluation under RNS-CKKS. The main three component of AutoFHE include: i) a multi-objective co-evolutionary (MOCoEv) search algorithm to maximize validation accuracy and minimize the number of bootstrapping operations, ii) a gradient-free search algorithm, R-CCDE, to optimize EvoReLU coefficients, and iii) polynomial-aware training (PAT) to fine-tune polynomial-only CNNs for one epoch to adapt trainable weights to EvoReLUs. The authors demonstrate the efficacy of AutoFHE on CIFAR-10 and CIFAR-100.

Strength:
- The paper provides a framework for secure CNN inference at the same time minimizing latency.
- Issues with FHE-MP-CNN is well identified: regards to use of  high order approximation for ReLUs
- The proposed techniques are sound
- Experiments and evaluations are complete

Weakness
- Issues in novelty for the search algorithm
- Applicability beyond CIFAR experiments
- More details in the AC-reviewer meeting notes

Among reviewers there was disagreement about novelty based on search algorithm vs secure inference. On top of that while some reviewers saw the paper clearly written, some saw issues in clarity in abstract and overall flow in the writing.


**Summary Of Ac-Reviewer Meeting:**

Since reviewer opinions were split among the reviewers, AC and the reviewers met during the discussion phase and tried to reach consensus.

Reviewer `Lx4T` pointed out in the eyes of secure inference from the field of security the idea is interesting and marked as solid progress. Reviewers pointed out empirical results are good and useful and may benefit the broader scientific community.  Also reviewers considered the current paper to be on the side of "good" security based paper among ICLR reviews they've done.

Reviewer `b7YQ` raised issue with co-optimization techniques which has been successfully used in ML community in neural architecture search (NAS) on high dimensional search spaces. Main reason for considering this is because for ML conferences such as ICLR, optimization method should be the important aspect of evaluation and does not seem to provide novelty.

Reviewer `b7YQ` agreed that scalability to high resolution can be a challenge and require novel research but still pointed out that showing applicability beyond CIFAR datasets to such as SVHN or Tiny Imagenet is possible. If theses analysis existed empirical evaluation and potential for practicality could be enough to support acceptance of the paper but the reviewer believed with current status the paper does not meet acceptance threshold.

Reviewer `Btk1` argued for marginal acceptance based on empirical findings which could benefit the community. Main argument that the experimental results could be useful. Originally, the reviewer had concerns over justification of design choices of the paper's methodology but clarified that the rebuttal helped ease the concern.

In the end, everyone agreed that ML methodology novelty is weak due to overlapping ideas already existing in the literature. Some saw the security implication and empirical results was solid enough to support marginal acceptance. The other reviewer saw that even similar empirical results exist in ML literature and wouldn't be able to support without evidence for broader applicability beyond CIFAR datasets.

The authors did try to address this issue using plaintext ImageNet evaluation during the discussion phase based on just showing polynomial approximation across depth. The reviewer was not satisfied with this evaluation, still arguing for low resolution and different dataset experiments.

Unfortunately, in the end reviewers did not reach consensus but was on the same footing that the paper is a borderline submission. Without a clear champion for the paper, reviewers' judgement that paper's main idea overlaps with existing literature, and that broad applicability is not clearly demonstrated empirically, the AC sees no strong evidence that the submission meets the acceptance bar, and thus propose rejection.